# HCRMP: A LLM-Hinted Contextual Reinforcement Learning Framework for Autonomous Driving

**Zhiwen Chen**
Tongji University
zhiwen_chen725@163.com

**Hanming Deng**
SenseTime Research
denghanming@sensetime.com

**Zhuoren Li**[*]
Tongji University
1911055@tongji.edu.cn

**Huanxi Wen**
Tongji University
2311385@tongji.edu.cn

**Guizhe Jin**
Tongji University
jgz13573016892@163.com

**Ran Yu**
Tongji University
ranyu@tongji.edu.cn

**Bo Leng**[*]
Tongji University
lengbo@tongji.edu.cn

## Abstract

Integrating the understanding and reasoning capabilities of Large Language Models (LLM) with the self-learning capabilities of Reinforcement Learning (RL) enables more reliable driving performance under complex driving conditions. There has been a lot of work exploring LLM-Dominated RL methods in the field of autonomous driving motion planning. These methods, which utilize LLM to directly generate policies or provide decisive instructions during policy learning of RL agent, are centrally characterized by an over-reliance on LLM outputs. However, LLM outputs are susceptible to hallucinations. Evaluations show that state-of-the-art LLM indicates a non-hallucination rate of only approximately 57.95% when assessed on essential driving-related tasks. Thus, in these methods, hallucinations from the LLM can directly jeopardize the performance of driving policies. This paper argues that *maintaining relative independence between the LLM and the RL* is vital for solving the hallucinations problem. Consequently, this paper is devoted to propose a novel **LLM-Hinted RL** paradigm. The LLM is used to generate semantic hints for state augmentation and policy optimization to assist RL agent in motion planning, while the RL agent counteracts potential erroneous semantic indications through policy learning to achieve excellent driving performance. Based on this paradigm, we propose the **HCRMP** (LLM-Hinted Contextual Reinforcement Learning Motion Planner) architecture, which is designed that includes ① Augmented Semantic Representation Module to extend state space. ② Contextual Stability Anchor Module enhances the reliability of multi-critic weight hints by utilizing information from the knowledge base. ③ Semantic Cache Module is employed to seamlessly integrate LLM low-frequency guidance with RL high-frequency control. Extensive experiments in CARLA validate its strong overall driving performance. HCRMP achieves a task success rate of up to 80.3% under diverse driving conditions with different traffic densities. Under safety-critical driving conditions, HCRMP significantly reduces the collision rate by 11.4%, which effectively improves the driving performance in complex scenarios.

---

[*]Corresponding authors.

39th Conference on Neural Information Processing Systems (NeurIPS 2025).

# 1  Introduction

Reinforcement learning (RL) is a method for learning optimal policies by maximizing expected returns through interactions with the environment. It has proven to be effective for solving complex decision-making problems and has attracted significant attention in various fields [1, 2, 3]. For the motion planning task in autonomous driving (AD), the RL agent can dynamically generate trajectories or control commands that follow the learned driving policy, based on the fused multimodal traffic features[4, 5, 6]. However, RL has limited understanding and reasoning in complex driving conditions and often fails to identify critical traffic features[7, 8], which can result in unreliable driving actions. In contrast, large language models (LLM) poss strong semantic understanding and common-sense reasoning abilities[9, 10], and have achieved significant advances in multitasking in recent years[11, 12, 13]. AD systems require the extensive common sense and high-level decision-making capabilities, which are exactly what large language models can provide[14, 15, 16, 17, 18]. Therefore, integrating LLM and RL within a unified AD system leverages the strengths of both approaches[19]. This integration enhances policy understanding and reasoning during self-learning[20], leading to more reliable and safer driving actions in complex driving conditions[21].

Despite the significant potential benefits of integrating LLM and RL for AD, effective integration remains a critical and unresolved challenge. Current methods primarily use RL agent to assist in the policy optimization of LLM [22, 23, 24] or employ LLM to directly instruct RL agent policy generation [25, 26, 27, 28, 29, 30, 31, 32]. In the former approach, the LLM outputs directly generate the driving policy, while in the latter, the LLM strongly provides decisive instructions during the policy learning process of the RL agent. Because both methods demonstrate strong reliance on the LLM, we refer to such methods as LLM-Dominated RL Methods.

However, LLM outputs are known to be susceptible to hallucinations [33, 34, 35, 36, 37], which can distort decision-making and compromise policy stability. The Gemini-2.5-Pro model, a state-of-the-art (SOTA) LLM [38], is evaluated on five key dimensions related to two essential driving capabilities: scenario understanding and action response, as illustrated in Figure 1 (a). The result shows a non-hallucination rate of only 57.95%, implying that over 40% of its outputs are prone to hallucinations. Further details are provided in the Appendix A.

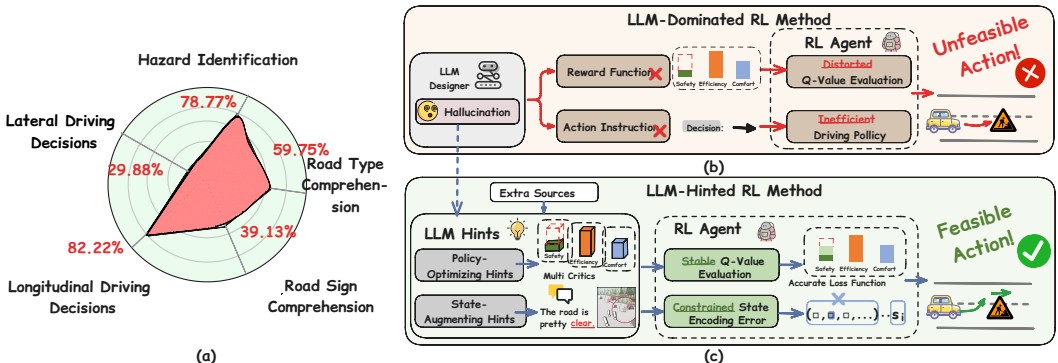

Figure 1: LLM performance evaluation and hallucination impact on LLM-RL methods. Figure (a) shows the SOTA LLM's non-hallucination rates across five driving tasks. Figure (b) illustrates the LLM-Dominated RL methods. LLM hallucinations directly distort the RL agent's Q-value estimation and degrade policy efficiency, leading to unfeasible driving actions. Figure (c) presents the LLM-Hinted RL method. LLM provides semantic hints to the RL agent instead of directly dictating decisions. The RL agent, through its own policy learning process, effectively buffers the negative impact of these hallucinations, thereby preventing unfeasible actions.

To clarify how these capabilities interface with RL, LLMs are typically integrated in two primary modes. In **LLM-Instructed RL Policy Generation**, the LLM primarily acts as an instructor, leveraging its scenario understanding capability. It interprets complex traffic environments (e.g., Hazard Identification, Road sign comprehension) and translates this high-level understanding into dynamically shaped reward functions for the RL agent. Conversely, in **RL-Assisted LLM Policy**

**Optimization**, the LLM is the core decision-maker, leveraging its action response capability to directly generate high-level driving policies (e.g., Longitudinal Driving Decisions or Lateral Driving Decisions), which the RL agent then assists in optimizing through environmental feedback. In both modes, the LLM's output is tightly coupled with the final driving decision. As shown in Figure 1 (b), when LLM hallucinations occur, these incorrect signals are directly propagated to the downstream decision-making process. This distorts the Q-value evaluation and compromises the safety and stability of the policy. The direct result is a significant degradation in overall driving performance. It is thus imperative for AD systems combining LLM and RL to mitigate the policy instability caused by LLM hallucinations.

This paper argues that *maintaining relative independence between the LLM and the RL* is an essential way to solve the hallucinations problem. The fundamental reason is that such separation preserves the RL agent's autonomy in decision-making and adaptation, while allowing the LLM to provide semantic hints as auxiliary inputs and intrinsic modulation. Accordingly, We propose an LLM-Hinted RL motion planning paradigm, as illustrated in Figure 1 (c). Even if the LLM outputs are unstable, the RL agent is able to counteract potential erroneous semantic indications through policy learning, avoiding the direct generation of unreasonable actions. At the same time, this separated structure preserves the utilization of LLM strengths in driving conditions comprehension and common-sense reasoning, enabling context-aware guidance for the RL agent in a way that maintains its fundamental self-optimization capabilities.

Based on the LLM-Hinted RL paradigm, we develop an architecture called HCRMP (LLM-**H**inted **C**ontextual **R**einforcement Learning **M**otion **P**lanner). The architecture comprises three key components: ① the Augmented Semantic Representation Module, which utilizes semantic hints from the LLM to extend the state space; ② the Contextual Stability Anchor Module, which leverages information from the structured knowledge base to improve the reliability of the weight hints that the LLM generates for each critic network; and ③ the Semantic Cache Module, which enables efficient and stable training through fixed-frequency hierarchical outputs and the historical context cache matching strategy. The LLM provides state-augmenting and policy-optimizing semantic hints as auxiliary inputs and intrinsic modulation to the RL agent, rather than directly controlling policy generation. Meanwhile, the RL agent autonomously executes motion planning, and the LLM and RL modules collaborate asynchronously at different temporal scales, ensuring training stability. In addition, extensive experiments in the CARLA simulator validate the effectiveness of our proposed HCRMP framework, highlighting its superior overall performance, particularly in demanding driving conditions. HCRMP achieves a task success rate of up to **80.3%** across diverse conditions with varied traffic densities. Critically, in safety-critical driving conditions, HCRMP achieves a significant **11.4%** reduction in the collision rate. The contributions of this study can be summarized as follows:

- We classify existing LLM-Dominated RL methods, clarify their strong reliance on LLM outputs, and highlight the problem that hallucinations from the LLM can degrade driving performance. To address these challenges, we propose the LLM-Hinted RL paradigm.

- We propose a novel motion planning architecture named HCRMP. By combining the semantic hints for state augmentation and policy optimization provided by LLM with the self-learning capabilities of RL, it significantly improves driving performance in diverse driving conditions.

- Extensive experiments in CARLA validate HCRMP's strong overall driving performance. HCRMP achieves a task success rate of up to **80.3%** under diverse driving conditions and, critically, reduces the collision rate by **11.4%** in safety-critical driving conditions.

## 2 Related Works

LLM-Dominated RL methods for AD motion planning fall into two main categories: RL-assisted LLM policy optimization and LLM-instructed RL policy generation. Additional related work is provided in Appendix B.

### 2.1 RL-Assisted LLM Policy Optimization

This type of methods typically convert conditions information into linguistic inputs to generate action instructions or probabilities distributions [39, 40, 41, 42, 43, 44]. RL agent enables the fine-tuning of LLM parameters using reward signals from the environment, optimizing its policy to maximize

cumulative rewards [45, 46]. Recently, the integrated paradigm of LLM and RL has demonstrated considerable potential for motion planning in AD. Existing studies primarily use LLM to generate trajectories or control commands, with RL submodules integrated for action suggestion generation or policy optimization. HighwayLLM [22] drives an LLM agent using meta-actions output by a pre-trained RL model, combining the current state and similar trajectories to generate specific actions. AlphaDrive [23], in contrast, leverages GRPO-based RL reward function to enhance the driving policy of its vision-language models (VLM). This enhancement enables the VLM to better adapt to dynamic driving conditions. While these methods can leverage the LLM strengths in complex scenario understanding and decision-making, their core risk is that any erroneous instructions generated by LLM can be directly mapped to unreasonable driving actions, which fundamentally threaten the driving safety.

## 2.2 LLM-Instructed RL Policy Generation

For the other type of method, RL agent is used to generate control commands. LLM serves as a sub-module to support the policy optimization. Specifically, LLM is utilized to provide intrinsic rewards, which is proven to improve the learning efficiency of RL [47, 48, 49, 50, 51]. For AD motion planning, LearningFlow [28] and Autoreward [29] utilize a closed-loop framework in which the LLM automatically generates reward signals to guide RL agent training. Similarly, Clip-RLdrive [30], LORD [31], and REvolve [32] directly rely on the LLM to generate reward values for policy learning. However, these methods are highly sensitive to the quality of LLM outputs and are therefore quite susceptible to the adverse effects of hallucinations. In contrast, our proposed method utilizes a *weakly coupled* integration of the LLM and RL. This significantly diminishes the system's vulnerability to fluctuations in the LLM outputs, preserving the LLM inherent semantic and reasoning advantages while ensuring RL maintains its fundamental self-optimization capabilities.

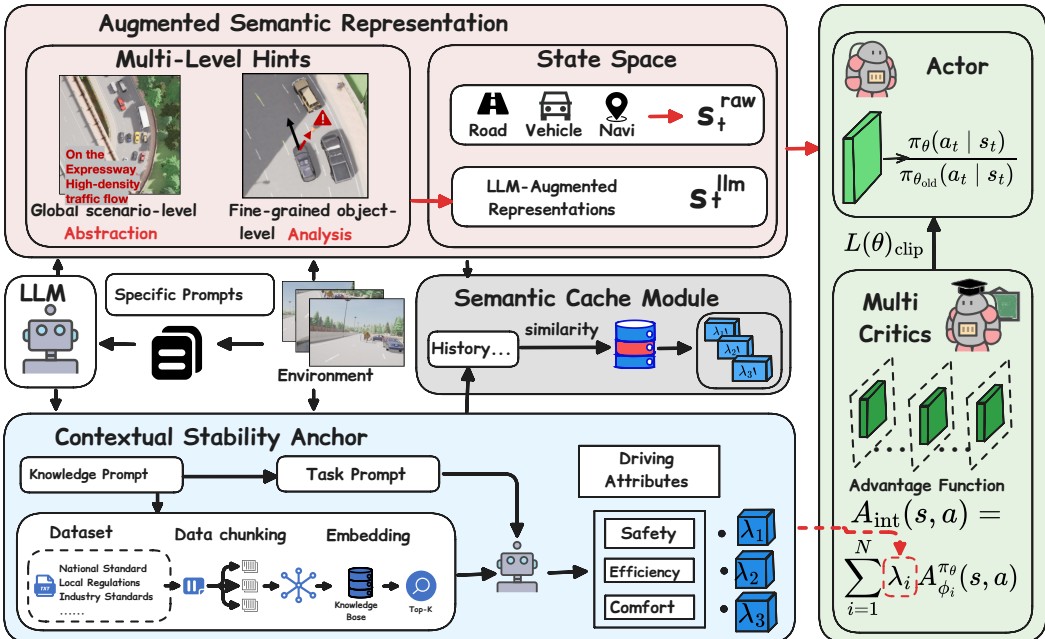

Figure 2: The framework of our proposed HCRMP. LLM acts in the Augmented Semantic Representation module to fetch information at the scenario level and object level, extending the state space. Meanwhile, LLM acts in the Contextual Stability Anchor module to generate reliable weights between multi critics, utilizing the knowledge base to mitigate the output fluctuations. When LLM fails to provide timely guidance, the Semantic Cache module replaces the missing weights by retrieving the most similar historical driving conditions. Hints from LLM are ultimately input to the RL agent's actor and multi-critic networks for optimal policy learning.

# 3 Methodology

This section outlines the proposed HCRMP framework. As illustrated in Figure 2, HCRMP comprises three key components: the Augmented Semantic Representation Module (Section 3.2), the Contextual Stability Anchor (Section 3.3), and the Semantic Cache Module (Section 3.4).

The Augmented Semantic Representation (ASR) module employs LLM for hierarchical scenario reasoning, from global abstraction to object-level analysis. The resulting multi-level semantic hints are encoded into compact vectors and integrated into the RL agent's state space to enhance driving condition awareness. The Contextual Stability Anchor (CSA) Module utilizes external knowledge sources, including traffic regulations and fundamental priors, to generate more reliable semantic hints. These hints then dynamically influence Q-value evaluation by modulating the impact of various driving attributes, thereby guiding the RL agent toward more effective decision-making. Furthermore, the Semantic Cache Module(SCM) addresses the temporal mismatch between the low-frequency semantic guidance from the LLM and the high-frequency control commands of the RL.

## 3.1 Problem Formulation

We formulate the AD task as a Markov Decision Process (MDP), represented by the tuple $(S, A, P, R, \gamma)$. Here, the action space $A$ consists of two continuous control variables: throttle/brake control commands and steering angle, each constrained within the normalized range $[-1, +1]$. The state space $S$ is composed of two components: the raw state characteristics $s_t^{\text{raw}}$ and the LLM-augmented semantic state characteristics $s_t^{\text{llm}}$. The transition function $P(s' \mid s, a)$ defines the probability distribution over next states, the reward function $R$ provides feedback signals reflecting the quality of actions, and $\gamma \in [0, 1]$ is a discount factor that trades off between immediate and future rewards. The agent employs a deep neural network (DNN) policy, denoted by $\pi_\theta$, where $\theta$ represents the learnable parameters. At each time step $t$, given the current state $s_t \in S$, formed by concatenating the raw sensory inputs $s_{raw}^t$ and the LLM-enhanced semantic embeddings $s_{llm}^t$, the policy samples a bounded continuous action $a_t \sim \pi_\theta(\cdot \mid s_t)$ from a Beta distribution, ensuring constrained control commands. Then, the agent executes action $a_t \in A$, receives an immediate reward $r_t \in R$, and transitions to the next state $s_{t+1}$ according to the probability $P$. The agent's objective is to discover an optimal policy $\pi^*$ through interaction with the environment that maximizes the expected discounted cumulative return in *Eq* 1.

$$\pi^* = \arg\max_\pi \mathbb{E}\left[\sum_{t=0}^{\infty} \gamma^t r(s_t, a_t)\right] \tag{1}$$

## 3.2 Augmented Semantic Representation

The LLM-based ASR module is designed to provide the RL agent with multi-level semantic encodings, which in turn augments its state space. The LLM performs hierarchical situational reasoning, transitioning from a global scenario-level abstraction to a fine-grained object-level analysis. Based on the scenario's topological configuration and traffic dynamics, the LLM executes semantic parsing to categorize road situations. Then, this categorized understanding is encoded into a 4-dimensional scenario-level vector. Concurrently, by analyzing the semantic characteristics of surrounding traffic participants, the LLM pinpoints critical agents. Following this identification, it quantifies their relative spatial configurations and maps them to discrete directional categories. These are then encoded as a 9-dimensional object-level vector. To ensure consistent representation, particularly in low-density traffic conditions, a semantic compensation mechanism can be implemented to preserve integrity.

At each time step $t$, the state space of the agent $S_t$ is constructed by combing heterogeneous feature sets, including raw sensory inputs $s_t^{\text{raw}}$ and LLM-enhanced semantic embeddings $s_t^{\text{llm}}$ - where $s_t^{\text{raw}} = (f_t^1, f_t^2, \ldots, f_t^J)$, and each feature frame $f_t^j = (I_t^j, T_t^j, E_t^j, \mathcal{N}_t^j)$ comprises multi-view RGB image features $I_t^j$, road topology features $T_t^j$, ego-vehicle dynamics $E_t^j$, and navigation path embeddings $\mathcal{N}_t^j$. The RL agent processes incoming multimodal inputs via a lightweight yet effective visual backbone (ShuffleNetV2 [52]) for spatial encoding and Gated Recurrent Units (GRU) for temporal semantic modeling. In this pipeline, the semantic representation $s_t^{\text{llm}}$ functions as an auxiliary information source, serving as semantic guidance to enhance situational awareness.

## 3.3 Contextual Stability Anchor

To satisfy the demands for adaptive trade-offs among driving attributes (e.g., safety, comfort, efficiency), we design a multi-critic framework based on Proximal Policy Optimization (PPO) [53]. The framework decouples the Q-value evaluation of multiple driving attributes and enable independent representation learning for distinct objectives. By formulating prompts, the LLM performs contextual analysis of dynamic traffic conditions and dynamically generates a set of adaptive weights $\{\lambda_i\}_{i=1}^N$, where $N$ is the number of attributes, $\lambda_i \in [0, 1]$, and $\sum_{i=1}^N \lambda_i = 1$, which are then used as part of an integrated advantage function $\hat{A}_{\text{int}}$ estimated via Generalized Advantage Estimation (GAE) in *Eq 2*

$$\hat{A}_{\text{int}}(s, a) = \sum_{i=1}^N \lambda_i A_i^{\pi_\theta}(s, a) \tag{2}$$

The clipping objective is defined in *Eq 3*, where the clip$(\cdot)$ function constrains the update range to stabilize training. $ratio_t(\theta)$ denotes the probability ratio between the current policy $\pi_\theta$ and the previous policy $\pi_{\theta_{\text{old}}}$.

$$L_{\text{clip}}(\theta) = \mathbb{E}_t \left[ \min \left( ratio_t(\theta) \hat{A}_{\text{int}}, \text{clip}(ratio_t(\theta), 1 - \varepsilon, 1 + \varepsilon) \hat{A}_{\text{int}} \right) \right] \tag{3}$$

To further enhance the stability of the LLM output dynamic weights, we propose a semantic anchoring module that utilizes Retrieval-Augmented Generation (RAG [54]) for stability optimization. The proposed module utilizes a semantic reference corpus. This corpus is compiled from authoritative sources such as national standards, local regulations, industry guidelines, and technical specifications [55, 56]. This module performs a text-to-text alignment: using a pretrained embedding model (`text-embedding-ada-002`), we embed both the real-time driving conditions query (Query) and the reference corpus (Knowledge) into high-dimensional vector representations. Semantic relevance between the query and corpus vectors is subsequently assessed using a FAISS-based similarity search [57]. A Top-3 selection strategy is employed to retrieve the most semantically aligned textual fragments. The retrieved passages are aggregated into a semantic context set, which is then provided as auxiliary input to the LLM. By anchoring weight generation to a knowledge base of traffic regulations and safety norms through RAG, this module mitigates weight drift, cutting LLM weight variance by 54.67% in complex driving (e.g., road construction) as detailed in Appendix C, and enhancing multi-attribute coordination. Importantly, the LLM-generated attribute weighting mechanism is implemented in a *weakly coupled* manner. This design leverages the LLM's strength in context-aware weighting while mitigating the potential adverse effects of hallucinated outputs on Q-value evaluation, thereby suppressing policy misguidance caused by hallucinations.

## 3.4 Semantic Cache Module

The inference latency of LLMs inherently limits the control frequency of existing LLM-Dominated RL methods. To address this problem, we propose a cooperative asynchronous training framework that coordinates low-frequency semantic planning by the LLM with the high-frequency control execution handled by the RL. However, this decoupling introduces a new challenge: the LLM can fail to deliver timely semantic outputs since unexpected system delays or interruptions.

To mitigate this, SCM compensates for such missing outputs by retrieving pertinent historical semantic representations. The SCM maintains a dedicated memory bank. This bank stores structured representations of past driving conditions, which encompass both scenario-level and object-level information. To manage this memory bank, the SCM employs a Least Recently Used (LRU) eviction policy to retain data from rare, critical scenarios. Crucially, it also includes the corresponding multi-critic weight vectors that are generated in previously similar conditions. When the LLM fails to return a valid semantic signal within a predefined time window, the SCM performs a rapid nearest-neighbor search over the memory bank using semantic embedding vectors. The module then identifies the historical entry most semantically aligned with the current driving context. It extracts the associated weight vector from this entry to serve as a temporary guidance signal for the LLM.

# 4 Experiments and Results

## 4.1 Experiment Setting

### 4.1.1 Driving Conditions

All experiments are conducted in Town 2 of the CARLA simulator [58]. Driving conditions include conventional conditions, such as overtaking and merging, as well as safety-critical conditions, such as trilemma and occluded pedestrian. To further assess the performance of the AD system, the traffic flow densities across three levels are established: low, medium, and high, as detailed in Appendix D.

### 4.1.2 Evaluation Metrics

We evaluate the driving policy using quantitative metrics for safety, efficiency, and comfort.

**Safety** is evaluated using two primary metrics: *Success Rate* (SR)—the percentage of episodes completed without major violations and *Collision Rate* (CR)—the proportion of episodes involving collisions.

**Efficiency** is evaluated by *Average Speed* (AS), *Total Distance* (TD) and *Time Steps* (TS), reflecting travel speed, distance covered, and task completion time, respectively.

**Comfort** is evaluated via *Speed Variance* (SV) and *Acceleration Variance* (AV), which reflects the smoothness and stability of driving behavior.

### 4.1.3 Baselines

We systematically compare the proposed method with the following methods:

- **Vanilla PPO** [53]: Directly train the policy using PPO in tasks as a basic RL baseline.
- **E2ECLA** [59]: Combine curriculum learning and RL, which learns end-to-end AD policies in CARLA by gradually increasing task difficulty without prior knowledge.
- **AutoReward** [29]: An RL method that iteratively refines LLM-generated rewards post-training.
- **VLM-RL** [25]: A method that integrates pre-trained VLM with RL. It generates semantic rewards through language objective comparison, replacing manually designed reward function.

## 4.2 Main Results

Table 1: Performance Comparison in Conventional Conditions under Different Traffic Densities

| Methods | Category | Condition | Low Density | | Medium Density | | High Density | |
| --- | --- | --- | --- | --- | --- | --- | --- | --- |
| | | | SR (%) | CR (%) | SR (%) | CR (%) | SR (%) | CR (%) |
| Vanilla PPO | RL | Overtaking | 80.0 | 20.0 | 69.0 | 31.0 | 62.0 | 38.0 |
| | | Merging | 89.0 | 11.0 | 75.0 | 25.0 | 68.0 | 32.0 |
| E2ECLA | RL | Overtaking | 60.0 | 40.0 | 56.0 | 44.0 | 38.0 | 62.0 |
| | | Merging | 56.0 | 44.0 | 54.0 | 46.0 | 42.0 | 58.0 |
| AutoReward (iter=0) | LLM-Dominated RL | Overtaking | 73.0 | 27.0 | 60.0 | 40.0 | 48.0 | 52.0 |
| | | Merging | 82.0 | 18.0 | 71.0 | 29.0 | 51.0 | 49.0 |
| AutoReward (iter=5) | LLM-Dominated RL | Overtaking | 85.0 | 15.0 | 76.0 | 24.0 | 70.0 | 30.0 |
| | | Merging | 94.0 | 6.0 | 85.0 | 15.0 | 71.0 | 29.0 |
| VLM-RL | LLM-Dominated RL | Overtaking | 54.0 | 45.0 | 52.0 | 48.0 | 50.0 | 50.0 |
| | | Merging | 56.0 | 44.0 | 53.0 | 47.0 | 50.0 | 50.0 |
| HCRMP | LLM-Hinted RL | Overtaking | 99.0 | 1.0 | 93.0 | 7.0 | 87.0 | 13.0 |
| | | Merging | 97.0 | 3.0 | 92.0 | 8.0 | 86.0 | 14.0 |

As illustrated in Table 1, SR and CR of various methods are compared in conventional conditions under different traffic densities. Results indicate that HCRMP matches or outperforms other baselines in different driving conditions, particularly in medium- and high-density conditions, where it achieves an average SR of 89.5%. By optimizing the multi-critic coordination strategy, the agent is able to adopt safer actions, significantly reducing the collision risk.

Additionally, HCRMP shows a relatively minor advantage over the baselines in low-density driving conditions, which is due to limited vehicle interactions. It reduces the difficulty of the driving task and restrict the full exploitation of the CSA's dynamic adjustment capabilities. We further conduct a systematic evaluation of HCRMP and three baselines-E2ECLA, VLM-RL, and Autoreward—under safety-critical scenarios, with the latter two being representative LLM-Dominated RL methods. The corresponding results are presented in Table 2.

Table 2: Performance Comparison in Safety-Critical Driving Conditions

| Methods | Condition | Traffic Density | SR (%) | CR (%) | AS (m/s) | TD (m) | TS (s) | SV (m/s) | AV (m/s$^2$) |
|---|---|---|---|---|---|---|---|---|---|
| E2ECLA | Occluded Pedestrian | Low | 36.0 | 64.0 | 8.22 | 36.99 | 47.04 | 4.29 | 1.49 |
| | | Medium | 37.0 | 63.0 | 7.19 | 31.53 | 39.72 | 3.02 | 2.62 |
| | | High | 31.0 | 69.0 | 7.87 | 33.86 | 37.04 | 3.25 | 1.52 |
| | Trilemma | Low | 36.0 | 64.0 | 6.99 | 52.82 | 48.38 | 3.01 | 2.89 |
| | | Medium | 34.0 | 66.0 | 6.99 | 40.64 | 50.28 | 2.18 | 2.70 |
| | | High | 38.0 | 62.0 | 8.68 | 48.96 | 46.44 | 4.59 | 2.89 |
| Autoreward(iter=5) | Occluded Pedestrian | Low | 70.0 | 30.0 | 6.30 | 28.07 | 35.05 | 1.40 | 2.96 |
| | | Medium | 31.0 | 69.0 | 7.79 | 27.42 | 55.78 | 1.34 | 3.16 |
| | | High | 24.0 | 76.0 | 7.37 | 31.26 | 53.65 | 2.15 | 3.21 |
| | Trilemma | Low | 60.0 | 40.0 | 5.60 | 24.91 | 34.3 | 2.45 | 2.68 |
| | | Medium | 58.0 | 42.0 | 6.16 | 31.22 | 33.9 | 3.05 | 3.33 |
| | | High | 32.0 | 68.0 | 7.05 | 25.84 | 55.27 | 1.90 | 2.95 |
| VLM-RL | Occluded Pedestrian | Low | 65.0 | 35.0 | 5.02 | 139.20 | 45.04 | 8.43 | 1.72 |
| | | Medium | 53.0 | 47.0 | 6.34 | 170.45 | 42.30 | 7.69 | 1.78 |
| | | High | 41.0 | 59.0 | 5.98 | 172.87 | 40.23 | 7.06 | 1.61 |
| | Trilemma | Low | 67.0 | 33.0 | 8.89 | 283.13 | 50.01 | 9.87 | 1.75 |
| | | Medium | 58.0 | 42.0 | 7.76 | 278.82 | 48.13 | 7.75 | 1.69 |
| | | High | 54.0 | 46.0 | 7.79 | 176.06 | 46.98 | 7.72 | 2.70 |
| HCRMP | Occluded Pedestrian | Low | 73.0 | 27.0 | 9.98 | 86.06 | 50.07 | 10.04 | 1.69 |
| | | Medium | 67.0 | 33.0 | 9.96 | 82.09 | 51.17 | 9.97 | 1.74 |
| | | High | 61.0 | 39.0 | 8.97 | 77.95 | 48.59 | 9.69 | 1.72 |
| | Trilemma | Low | 75.0 | 28.0 | 10.24 | 88.20 | 51.58 | 9.57 | 1.44 |
| | | Medium | 69.0 | 31.0 | 10.08 | 79.91 | 49.72 | 10.14 | 1.23 |
| | | High | 64.0 | 36.0 | 9.94 | 77.34 | 47.13 | 9.96 | 1.72 |

E2ECLA exhibits an AV exceeding 2.5 m/s² under certain conditions, which can cause discomfort to the passenger [60]. This phenomenon can be primarily attributed to E2ECLA's failure to adequately consider vehicle acceleration. Consequently, the system prioritizes rapid maneuvers over comfort. AutoReward utilizes LLM to construct the reward function based on the analysis of the scenario. Similarly, unreasonable values of acceleration changes occur due to the LLM's emphasis on maneuverability at the expense of comfort considerations in safety-critical scenarios. VLM-RL, leveraging efficient navigation approaches, achieves a notably high TD; however, its average CR in safety-critical conditions reaches 43.7%, indicating a significant safety concern.

The proposed HCRMP demonstrates a well-balanced performance across all metrics, particularly in high-density conditions, where it achieves a SR of 62.5%, a CR of 37.5%, and an AV of 1.72 m/s². This comprehensive performance is mainly due to the integration of ASR and CSA. The former

pursues efficiency and comfort in low-density traffic flows, while prioritizing safety in medium-and high-density traffic conditions. The latter enhances the system's ability to understand complex environments by extending the state space.

HCRMP prioritizes ensuring safety in immediate driving tasks. This focus, however, means its global adaptation to the map's inherent static path features may be less developed, which consequently results in a lower traveled distance compared to VLM-RL. Through the synergistic effect of CSA and ASR, HCRMP achieves an effective balance of safety, efficiency, and comfort in safety-critical conditions, with particularly outstanding performance in medium- and high-density conditions.

### 4.3 Ablation Study

Table 3 presents the results of the ablation study in the medium-density trilemma, evaluating the performance of the HCRMP with the removal of different modules: HCRMP without ASR, HCRMP without CSA, and HCRMP with ASR. In the HCRMP w/o CSA configuration, the multi-critic weights are set to fixed equal values to create a clear baseline against the dynamic weighting provided by CSA, thus isolating and highlighting the CSA module's contribution.

Table 3: Ablation Study Results

| Model | SR(%) | CR(%) | AS (m/s) | TD (m) | TS (s) | SV (m/s) | AV (m/s$^2$) | Collision per Mile |
|---|---|---|---|---|---|---|---|---|
| HCRMP w/o ASR | 40 | 60 | 7.52 | 44.34 | 47.74 | 6.94 | 2.95 | 21.65 |
| HCRMP w/o CSA | 48 | 52 | 6.01 | 29.82 | 54.44 | 2.89 | 2.63 | 27.90 |
| HCRMP w/ ASR | 54 | 46 | 7.27 | 29.04 | 34.64 | 5.89 | 2.26 | 25.34 |

The SR of HCRMP without ASR drops to a mere 40.0%, indicating that the absence of ASR significantly diminishes the system's ability to comprehend complex environments, resulting in information deficits that increase collision risks. In contrast, HCRMP with ASR, achieves an SR of 54.0%, underscoring the critical role of ASR in enhancing situational awareness. By leveraging the LLM to interpret the current driving conditions, ASR expands the state space, strengthening the system's awareness of surrounding driving risks and thereby improving the safety of its decision-making tasks.

Furthermore, HCRMP without CSA exhibits an SR of 48.0%, which is even lower than that of HCRMP with ASR. Specifically, CSA enhances contextual stability through a knowledge base by dynamically constraining the priority weights within the multi-critic framework, effectively mitigating excessive fluctuations during the training process. Without CSA, the multi-critic system experiences pronounced instability in weight adjustments, hindering convergence toward an optimal policy. This instability directly undermines the system's decision-making consistency and safety in the medium-density trilemma, consequently leading to a substantial decline in SR.

Figure 3 illustrates the performance differences between HCRMP variants through dynamic reward trends. The reward curve for HCRMP without CSA exhibits pronounced fluctuations, indicating that, in the absence of CSA constraints, the system struggles to stabilize reward values during evaluating, reflecting inherent instability in policy optimization. In contrast, the HCRMP variant equipped with CSA demonstrates a smoother upward trend in its reward curve.

## 5  Conclusion

Current LLM-Dominated RL approaches for AD motion planning heavily depend on LLM outputs, making them vulnerable to hallucinations that can compromise policy reliability and lead to unsafe behavior. We propose a LLM-Hinted RL motion planning paradigm and the corresponding HCRMP framework, aiming to preserve the relative independence between LLM and RL. The framework mitigates the impact of LLM hallucinations, while still preserving the strengths of LLM in semantic understanding and high-level decision-making while ensuring RL maintains its fundamental self-optimization capabilities. The HCRMP architecture comprises three key components. First, the ① **Augmented Semantic Representation Module** refines the state space via semantic guidance.

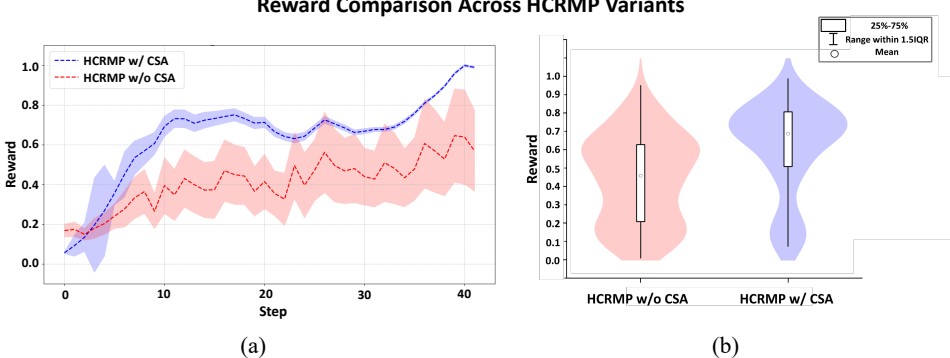

Figure 3: HCRMP variants rewards: dynamic trends and statistical distributions. Figure (a) visualizes that the dynamic reward curve for HCRMP without CSA exhibits significant fluctuations, indicative of performance instability, while the curve for HCRMP with CSA shows smaller fluctuations and more stable performance. Figure (b) illustrates that the overall reward distribution for the variant without CSA is wider and more pronounced in lower reward regions, whereas the CSA variant's rewards are more concentrated in higher value ranges with a more prominent peak.

Second, the ② **Contextual Stability Anchor Module** enhances the reliability of LLM-provided multi-critic weight hints through retrieval-augmented semantic anchoring based on information from the structured knowledge base. Finally, the ③ **Semantic Cache Module** primarily improves training efficiency by asynchronous decoupling low-frequency LLM reasoning from high-frequency RL execution. To handle delayed LLM outputs, it employs a historical semantic cache matching strategy as a fallback. Extensive experiments in CARLA validate HCRMP's strong overall driving performance. HCRMP has a high task success rate of 80.3% under diverse driving conditions with different traffic densities. Especially, it achieves a 11.4% reduction in collision rate across a range of complex driving conditions. HCRMP provides a promising framework for RL motion planning for AD with integrated LLM.

## 6 Acknowledgments

This work was supported in part by the National Natural Science Foundation of China under Grant 52522219 and 52232015, and in part by the Fundamental Research Funds for the Central Universities under Grant 22120230311.

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

# Appendix

# Table of Contents

# A    LLM Evaluation: Driving Scenario Understanding and Response

## A.1    Evaluation Setup

### A.1.1    Driving Conditions Specification

Driving conditions encompass both conventional scenarios—such as overtaking and merging—and safety-critical situations [1], as illustrated in Figure 1.

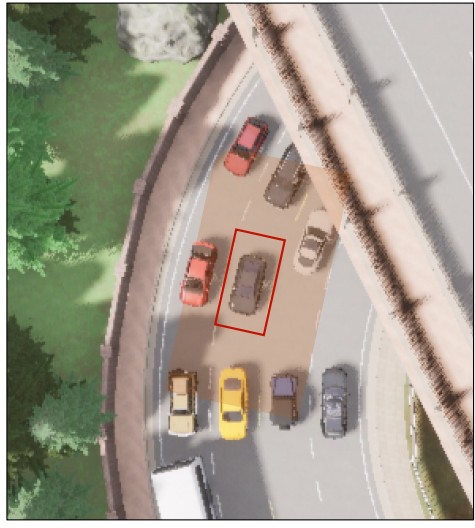
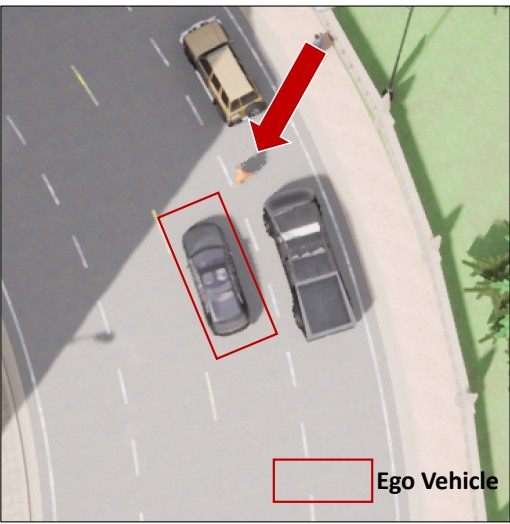

(a) Trilemma                                     (b) Occluded Pedestrian

Figure 1: Safety-Critical Driving Conditions: Trilemma and Occluded Pedestrian

The **Trilemma** driving condition refers to a situation where the ego vehicle (indicated by the red box) is surrounded by other vehicles on all sides. In this case, the ego vehicle must balance among three competing objectives: speed optimization, safe distance maintenance, and the need to change lanes. The **Occluded Pedestrian** driving condition is defined as a pedestrian suddenly stepping out from a blind spot and intersecting the path of the ego vehicle.

**The driving conditions used in this part of the experiment are consistent with those used in the other experiments in this paper.**

### A.1.2    Task Definition

Each task systematically evaluates different capabilities of LLM in the context of autonomous driving(AD). We categorize these tasks into two main groups: Scenario Understanding and Action Response. The former focuses on the recognition and interpretation of driving conditions, while the latter assesses the model's ability to respond and make decisions in various driving situations.

**Scenario Understanding:**

- **Hazard Identification**: the ability to accurately identify potential hazards in diverse driving conditions.
- **Road Type Comprehension**: the ability to correctly interpret and respond to different road types.
- **Road Sign Comprehension**: the ability to correctly interpret and respond to traffic signs.

**Action Response:**

- **Longitudinal Driving Decisions**: The ability to make appropriate decisions related to ego-vehicle speed.

- **Lateral Driving Decisions**: The ability to make appropriate decisions related to lane changes and steering.

### A.1.3   Evaluation Dimensions

The primary metric used is the non-hallucination rate, which quantifies the accuracy and factual correctness of the LLM outputs.

**Non-Hallucination Rate**

For each dimension, the LLM output is evaluated for the presence of hallucinations. A hallucination is defined as any generated content that is factually incorrect, inconsistent with the driving condition, or irrelevant to the query. The non-hallucination rate is calculated as:

$$\text{Non-Hallucination Rate} = \frac{\text{Number of Non-Hallucinated Outputs}}{\text{Total Number of Outputs}} \times 100\%$$

### A.2   Test Question Examples

We design a set of test questions to evaluate different LLM across the five defined tasks. Each test problem consists of a textual description of a driving condition, along with a specific query or instruction. The example problems are shown in Fig. 2 and aims to comprehensively evaluate the various capabilities of LLM in driving scenario understanding and response generation .

**Hazard Identification Example :**

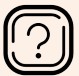 You're a self-driving car, What do you think is the biggest influence on you in your neighborhood right now?

**Road Type Comprehension Example :**

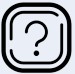 You're a self-driving car, where do you think you're traveling right now?

**Road Sign Comprehension Example :**

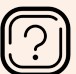 You're a self-driving car, what do you think are the traffic signs ahead?

**Longitudinal Driving Decisions Example :**

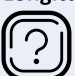 You're a self-driving car, What should your current longitudinal driving behavior look like?

**Lateral Driving Decisions Example :**

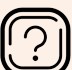 You're a self-driving car, What should your current lateral driving behavior look like?

Figure 2: Sample test question for each task

### A.3   Comparison Methods

- **Gemini-2.5-Pro** [2]: On March 21, 2025, Google DeepMind officially launched Gemini-2.5-Pro , its latest flagship model. Hailed as Google's 'most intelligent AI model' to date, it marks revolutionary breakthroughs in reasoning, context understanding, and multimodal processing.

- **Gpt-4o** [3]: A language model released for ChatGPT, offers real-time reasoning across audio, visual, and text inputs. It supports 50 different languages with improved speed and quality.

- **Deepseek-r1** [4]: An AI model developed by Chinese artificial intelligence startup DeepSeek. Post-trained with reinforcement learning, it's designed to enhance reasoning capabilities, particularly excelling at complex tasks such as mathematics, code, and natural language reasoning.

- **Qwen-Turbo** [5]: Alibaba has launched a Qwen model on its Alibaba Cloud Bailian platform, featuring a significantly increased context length from 128k tokens to 1M tokens. This is equivalent to approximately one million English words or one and a half million Chinese characters.

- **Llama-3.3-70B-Instruct** [6]: Developed by Meta, it is the LLM with 70 billion parameters. It's specifically designed for multilingual dialogue scenarios and has been optimized through Supervised Fine-Tuning (SFT) and Reinforcement Learning from Human Feedback (RLHF), enabling it to excel at natural language processing tasks such as text generation.

## A.4 Conclusions

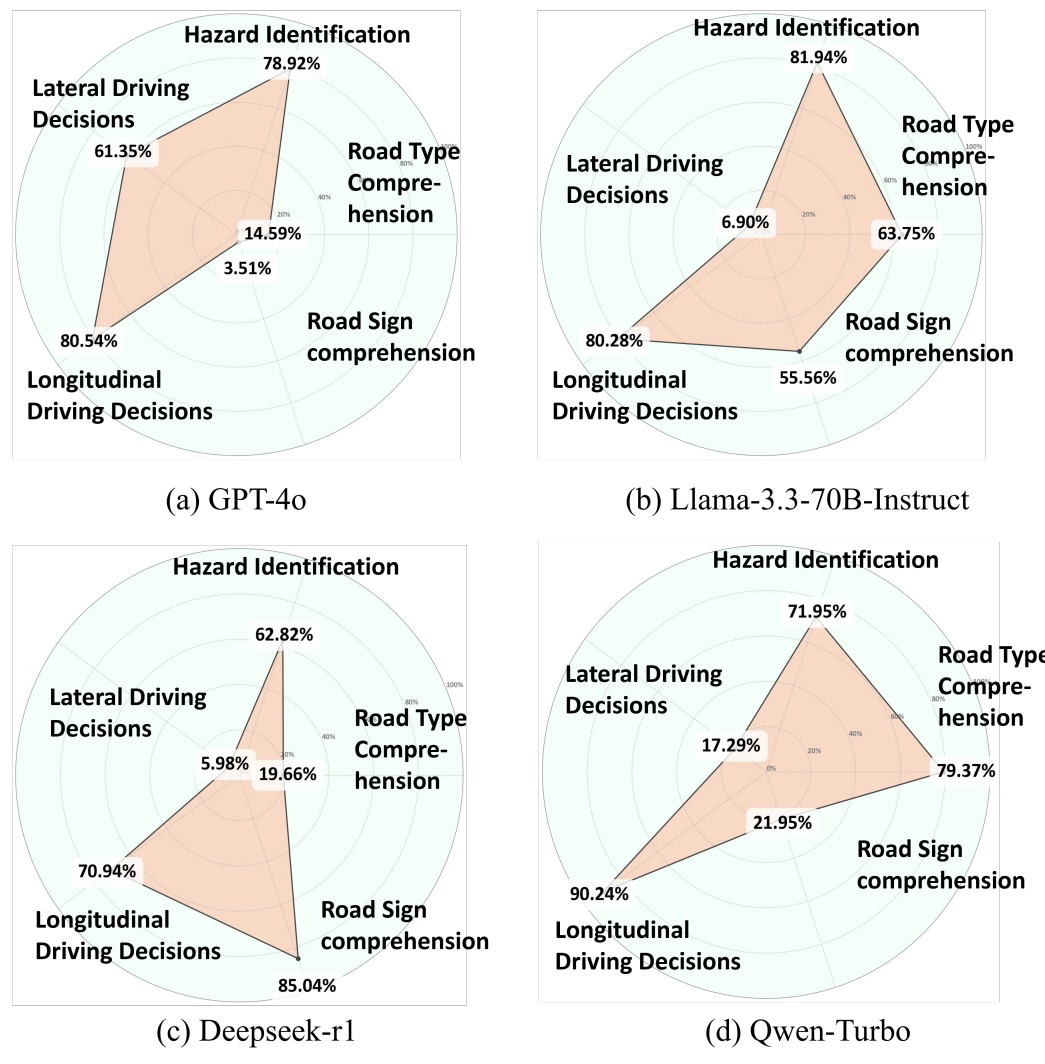

(a) GPT-4o        (b) Llama-3.3-70B-Instruct

(c) Deepseek-r1        (d) Qwen-Turbo

Figure 3: Non-Hallucination Rates of Different LLM Across Driving Tasks

Regarding the performance evaluation of LLM in autonomous driving tasks, results indicate that their performance is relatively strong in Longitudinal Driving Decisions and Hazard Identification, exhibiting higher non-hallucination rates. However, for tasks involving complex and precise operations such as Lateral Driving Decisions, the non-hallucination rates across all evaluated models are generally low, suggesting a significant hallucination rate in this area. Furthermore, apart from Deepseek-r1 which demonstrates prominent performance in Road Sign Comprehension, other models generally show deficiencies in this dimension. This highlights the inherent limitations of LLMs in accurate physical world perception and dynamic decision-making, which remain crucial challenges for their application in autonomous driving.

# B  Addition Related Work

## B.1  Taxonomy of LLM-RL Integration Paradigms for Autonomous Driving

To ensure the focus of our discussion in the main paper, we concentrate on studying the methods **most relevant to our work** based on the core mechanism of "how an LLM's output influences and integrates into the RL decision loop." This section provides a comprehensive taxonomy of LLM-RL integration paradigms to address the broader landscape of existing methods.

### B.1.1  Overview of LLM-RL Integration Paradigms

Table 1 presents a taxonomy of mainstream paradigms for LLM-instructed RL in autonomous driving, categorized by the LLM's primary role and key characteristics.

Table 1: Taxonomy of LLM-RL Integration Paradigms for Autonomous Driving

| Papers & Venues | Policy Optimization | Common-Sense Reasoning | High-level Policy | Scenario Generation | Human-in-Loop |
|---|:---:|:---:|:---:|:---:|:---:|
| ***LLM as Reward Designer*** | | | | | |
| LearningFlow (arXiv 2025) | ✓ | ✓ | | | |
| LORD (WACV 2025) | ✓ | ✓ | | | |
| AutoReward (used in main) | ✓ | ✓ | | | |
| ***LLM as Decision Provider*** | | | | | |
| TeLL-Drive (arXiv 2025) | ✓ | ✓ | ✓ | | |
| LaViPlan (arXiv 2025) | ✓ | ✓ | ✓ | | |
| ***LLM as Language Translator*** | | | | | |
| DriveGPT4 (ICLR 2024) | ✓ | | | | ✓ |
| Human-Centric AD (arXiv 2025) | ✓ | | | | ✓ |
| ***LLM as Scenario Generator*** | | | | | |
| ChatScene (CVPR 2024) | | ✓ | | ✓ | |
| CRITICAL (arXiv 2024) | | ✓ | | ✓ | |
| ***LLM as Hinter*** | | | | | |
| **HCRMP (Ours)** | ✓ | ✓ | | | |

### B.1.2  Rationale for Inclusion and Exclusion in Main Paper Discussion

We clarify our reasoning for including or excluding these paradigms from our core discussion in the main paper:

#### B.1.2.1  LLM as a Reward Designer
This paradigm fully leverages the LLM's powerful common-sense reasoning capabilities for collaborative policy optimization. The LLM analyzes driving scenarios and generates reward signals to guide RL agent training. Methods such as LearningFlow, LORD, and AutoReward fall into this category. As this direction directly aligns with our research focus on how LLM outputs influence RL policy learning, it is a key focus of our discussion in Section 2 (Related Work) of the main paper.

**Key Characteristic:** LLM outputs (reward functions or reward values) have a *decisive influence* on the RL agent's optimization objective, making these methods highly sensitive to LLM hallucinations.

#### B.1.2.2 LLM as a Decision Provider

This paradigm positions the LLM as a high-level decision-maker that outputs semantic policies (e.g., "turn left," "slow down"), which are then translated into specific vehicle control signals by an RL module or rule-based planner. Examples include TeLL-Drive and LaViPlan. This approach is fundamentally different from our end-to-end RL decision mechanism, where the RL agent directly generates low-level control commands (throttle, steering). Therefore, this paradigm is not a core topic of our main paper discussion.

**Key Difference:** The RL component acts as an *executor* rather than a *learner*, which represents a different technical direction from our work.

#### B.1.2.3 LLM as a Scenario Generator

This paradigm uses the LLM to enhance the training environment by generating diverse or safety-critical scenarios (e.g., ChatScene, CRITICAL). The LLM does not participate in policy optimization during training or inference. As this is focused on data augmentation rather than policy learning, it represents a different technical direction and is outside the scope of our related work.

**Key Difference:** LLM contributes to *training data diversity* rather than *policy optimization guidance*.

#### B.1.2.4 LLM as a Language Translator

This paradigm focuses on parsing human language commands (e.g., "drive faster," "turn right at the next intersection") and translating them into actionable instructions for the AD system, often in human-in-the-loop settings (e.g., DriveGPT4). Since our work does not process such natural language commands and focuses on autonomous policy learning without human intervention, we consider its relevance to be limited.

**Key Difference:** This paradigm addresses *human-vehicle interaction* rather than *autonomous policy optimization*.

### B.1.3 Positioning of Our LLM-Hinted RL Paradigm

Our proposed **LLM-Hinted RL** paradigm introduces a novel integration mechanism that differs from the *LLM as Reward Designer* category in a fundamental way:

- **LLM as Reward Designer (LLM-Dominated):** The LLM's outputs (reward functions or reward values) *dominate* the RL optimization objective. Any hallucination directly corrupts the training signal, leading to unsafe policies.
- **LLM as Hinter (LLM-Hinted, Ours):** The LLM provides *semantic hints* (state augmentation and multi-critic weight modulation) that assist but do not dominate the RL agent's policy learning. The RL agent retains its self-optimization capability and can buffer LLM hallucinations through policy learning and environmental feedback.

This *weakly coupled* integration mechanism is the core contribution of our work, distinguishing it from existing LLM-Dominated RL methods.

### B.2 Motion Planning in Autonomous Driving

The core task of motion planning in autonomous driving is to rapidly generate safe and robust local trajectories or motion commands that guide the vehicle to effectively avoid obstacles and operate smoothly in complex dynamic environments [7]. As a key component for achieving safe and efficient autonomous mobility, it has long been a highly active and closely studied area of research.

Current motion planning methods are mainly divided into two categories: pipeline planning and end-to-end planning. The traditional pipeline planning method, also known as the rule-based planning method, is composed of multiple interrelated modules. These modules—such as perception, localization, planning, and control—are designed and developed independently. Graph search planning algorithms determine a path between a start point and a goal point by performing iterative searches based on the environmental map and obstacle information. Common graph search algorithms include Dijkstra's algorithm, the A* algorithm, and the Hybrid A* algorithm. While Dijkstra's algorithm [8] can find the shortest path between two points, it lacks goal-directed efficiency and becomes computationally expensive in long-range searches. To address this limitation, Stanford University developed

the A* algorithm [9] in 1968, which significantly improves efficiency through the use of well-designed heuristic functions. However, A* is mainly suited for static environments and does not adequately account for the motion constraints of moving vehicles. In 2008, Stanford University introduced the Hybrid A* algorithm [10], which incorporates kinematic constraints, thereby enhancing the practicality and applicability of path planning in real-world driving scenarios. Esposto et al. [11] combined the Hybrid A* and classical A* algorithms to propose a path planning method based on Reeds-Shepp curves, which not only accommodates the kinematic characteristics of vehicles but also improves planning speed.

However, end-to-end planning methods have become a focal point of current research due to their superior adaptability and efficiency, leveraging artificial intelligence to directly map raw perception data to control commands. Representative approaches include: **Behavior Cloning**, which uses supervised learning to mimic expert trajectories and quickly generate reliable driving policies. **Reinforcement Learning**, which optimizes dynamic decision-making through interaction with the environment, enabling adaptation to complex scenarios.

Behavioral Cloning (BC) is a primary imitation learning approach in autonomous driving, where the agent learns to replicate expert behavior by training a classifier or regressor on demonstration trajectories. As a passive method, it assumes that state-action pairs in the demonstrations are independent and learns the target policy purely through observation of complete expert executions. Early BC applications in driving [12–14], used end-to-end neural networks to map camera inputs directly to control commands. To improve performance in complex urban settings, later work introduced enhancements such as multi-sensor inputs [15, 16], auxiliary learning tasks [17, 18], and more sophisticated expert demonstrations [19].

To reduce reliance on labeled data, some researchers have turned to reinforcement learning (RL) for autonomous decision-making. Unlike imitation learning, RL agents learn policies by interacting with the environment and maximizing cumulative rewards through trial and error. Over time, the agent refines its policy to achieve optimal performance based on feedback from the environment. RL has demonstrated success in learning lane following on a real vehicle in low-traffic conditions [20]. Saxena et al. [21] use the Proximal Policy Optimization (PPO) algorithm to learn a control policy in continuous motion planning. Their model implicitly accounts for interactions with surrounding vehicles to prevent collisions and improve ride comfort.

## C   Methodology Details

### C.1   Augmented Semantic Representation Prompts

---

**ASR Module Prompt: Global Scenario-Level Abstraction**

You are an advanced AI assistant for an autonomous driving system. Your task is to analyze the provided driving condition data and extract key scenario-level semantic information.
**Given Scene Data:**
- Road topology features: `{road_topology_description}`
- Current traffic dynamics (overall flow, presence of traffic signals/signs): `{traffic_dynamics_description}`
- Ego-vehicle state (position, speed): `{ego_vehicle_state}`

**Instructions:** Based on the provided driving condition data, please provide a concise answer for each of the following aspects. This information will be used to generate a 4-dimensional semantic vector representing the global driving context.

1. **Road Category:** Classify the current road type. (Examples: Highway, Rural Lane, Urban Road)

2. **Traffic Density:** Describe the prevailing traffic density. (Examples: Low, Medium, High)

**Output Format:** For each numbered point above, provide a short descriptive phrase. Example: 1. Road Category: Urban Road 2. Traffic Density: Medium

---

> **ASR Module Prompt: Fine-Grained Object-Level Analysis**
>
> You are an advanced AI assistant for an autonomous driving system. Your task is to analyze the provided driving condition data, focusing on surrounding traffic participants and obstacles, to extract critical object-level semantic information.
> **Given Scene Data:**
> - Detected surrounding traffic participants (vehicles, pedestrians, cyclists): `{list_of_detected_participants_with_type_position_velocity}`
> - Detected static obstacles (roadblocks, debris): `{detected_static_obstacles}`
> - Ego-vehicle state (position, speed, current lane, heading): `{ego_vehicle_state}`
>
> **Instructions:** Based on the provided scene data, identify up to 3 of the most critical (highest risk or most influential on ego-vehicle's decisions) traffic participants or obstacles. For each identified critical entity, provide the following:
> - **Entity Type:** (e.g., Car, Truck, Bus, Motorcycle, Pedestrian, Cyclist)
> - **Relative Direction:** Its primary direction relative to the ego vehicle. Choose from: Front, Front-Left, Front-Right, Left, Right, Rear-Left, Rear, Rear-Right.
>
> This information will be used to generate a 9-dimensional semantic vector representing the object-level context. Focus on conciseness and relevance for immediate driving decisions.
> **Output Format:** List each critical entity as a separate item. Example: - Entity 1: Type: Car, Direction: Front-Left - Entity 2: Type: Pedestrian, Direction: Right - Entity 3: Type: Truck, Direction: Front

## C.2   Contextual Stability Anchor

### C.2.1   Prompts

> **Prompt for Autonomous Driving Attribute Weights Generation**
>
> **Background:** I need to use a three-element vector to represent the weighting characteristics of safety, comfort, and efficiency in autonomous driving. Please analyze the current driving scene based on the data read from the JSON file. The ego vehicle's coordinates are: (`{ego_position_x}`, `{ego_position_y}`) The ego vehicle's velocity is: (`{ego_velocity_x}`, `{ego_velocity_y}`) Surrounding environment information is: `{surrounding_info}`, `{scene_flag}`
>
> **The question is:** `{question}`. Based on my requirements and `{rag_output}`, please generate a three-element vector for me. The three elements have the following requirements:
> - The three elements represent: Safety, Comfort, Efficiency.
> - The sum of the three elements is 1.
> - All three elements should be retained to 2 decimal places.
>
> Please determine the specific values based on my requirements and only output this vector without any other redundant content.

### C.2.2   Overview of Weight Distributions

By anchoring the weight generation process to a norm-constrained semantic source, this module significantly mitigates the risk of weight drift. To empirically validate this, we conducted experiments within the same complex driving condition, specifically a road construction environment, comparing the LLM-generated critics weights with and without the integration of the Contextual Stability Anchor (CSA) module. As illustrated in Figure 4, this module successfully reduces the variance of LLM-generated critics weights by approximately **54.67%** within the same driving condition, thereby enhancing the consistency of multi-attribute coordination.

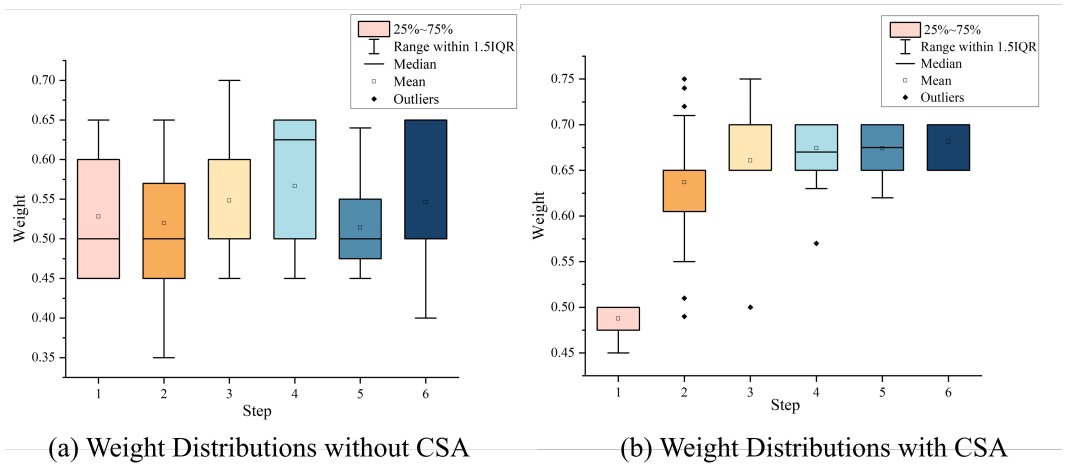

(a) Weight Distributions without CSA    (b) Weight Distributions with CSA

Figure 4: Comparison of LLM-Generated Weight Distributions With and Without CSA

Building upon the previous findings that demonstrated CSA's ability to stabilize LLM-generated weights within a single driving condition, we further investigated its impact across diverse driving conditions. Figure 5 presents the Kernel Density Estimation (KDE) of the LLM's output safety weights across various scenarios. In the left panel, representing the system without the CSA module, the boundary between safety weights for safe and dangerous scenarios is indistinct, indicating that the LLM struggles to consistently differentiate risk levels. Conversely, the right panel, which incorporates the CSA module, clearly illustrates a distinct separation in safety-critic weight distributions between safe and dangerous driving conditions. This evident demarcation highlights CSA's critical role in guiding the LLM to generate more context-aware and robust safety weight assignments, thereby enabling a clearer distinction between varying levels of risk and enhancing the reliability of safety-critical decision-making.

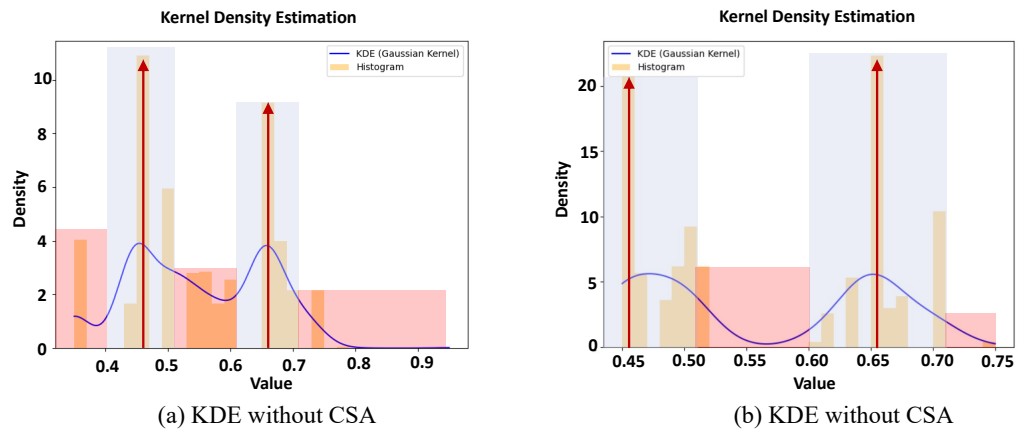

(a) KDE without CSA       (b) KDE without CSA

Figure 5: Kernel Density Estimation of LLM-Generated Safety-Critic Weights Across Diverse Driving Scenarios

### C.2.3 Comparison with Learnable Alternatives

To justify the design choice of using the LLM-based CSA module over a purely learnable alternative, we conducted a supplementary experiment comparing our approach with an end-to-end learnable network for critic weight generation.

### C.2.3.1 Experimental Setup

We replaced the CSA module with an end-to-end network that employs a 4-head self-attention mechanism followed by a multi-layer perceptron (MLP) to generate critic weights. This attention-based network is trained jointly with the PPO algorithm to learn the weight generation directly from driving data. The comparison was conducted in the challenging medium-density "Trilemma" scenario, which requires balancing speed optimization, safe distance maintenance, and lane change decisions.

### C.2.3.2 Results

Table 2 presents the performance comparison between the two approaches.

Table 2: Performance Comparison: CSA vs. Attention-Based Weight Generation

| Model | SR (%) | CR (%) |
|---|---|---|
| HCRMP with Attention Network | 57.0 | 43.0 |
| HCRMP with CSA (Ours) | 69.0 | 31.0 |

### C.2.3.3 Analysis

The experimental results indicate that the attention mechanism alone struggles to learn robust weight dynamics in the complex feature space of autonomous driving, which requires understanding multi-source inputs and dynamically adapting to diverse scenarios. We attribute this performance gap to the following fundamental differences:

- **Nature of the Task**: Determining appropriate safety weights is fundamentally a *reasoning problem* involving traffic regulations, risk assessment, and causal understanding, rather than a simple pattern recognition or correlation task. For instance, deciding that safety should be prioritized in a construction zone requires understanding traffic rules and consequences, not just recognizing visual patterns.

- **Knowledge Grounding**: Our CSA module leverages the LLM's pre-trained common-sense reasoning capabilities and explicitly anchors weight generation to a structured knowledge base (via RAG). This grounding ensures that the generated weights are consistent with safe driving regulations and established safety principles. In contrast, the attention-based network must learn these complex reasoning patterns from scratch using only driving trajectories, which is substantially more challenging and data-inefficient.

- **Stability and Interpretability**: The knowledge-anchored approach provides inherent stability (54.67% variance reduction as shown in Section C.2.2) and interpretability - the weights are grounded in retrievable knowledge sources. The end-to-end learned weights, while adaptive, lack this explicit grounding and exhibit higher variance across similar scenarios, leading to less consistent safety prioritization.

This comparison validates our design choice: for the safety-critical task of weight generation in autonomous driving, leveraging LLM reasoning with knowledge anchoring (the "hinting" mechanism) is more effective than relying solely on end-to-end learning from driving data.

### C.3 System Flexibility and Prompt Robustness

A critical concern for LLM-based systems is their flexibility and adaptability to scenarios not explicitly covered by predefined prompts. To address this, we conducted a comprehensive evaluation of HCRMP's robustness to prompt variations, testing whether the system maintains stable performance across different prompt formulations.

### C.3.1 Experimental Setup

We selected three representative driving scenarios under medium traffic density: "Overtaking," "Merging," and "Trilemma." For each scenario, we designed three prompt sets with different levels of detail and wording:

- **Prompt A (Detailed Version)**: An expanded version of the original prompt with more elaborate instructions and contextual descriptions.

- **Prompt B (Concise Version)**: A simplified version with condensed wording and minimal instructions.

- **Prompt C (Original Version)**: The baseline prompt used in the main experiments (as shown in Sections C.1 and C.2.1).

The full text of all three prompt versions is provided in Section C.3.3 below. We evaluated performance across three key metrics: Success Rate (SR), Collision Rate (CR), and the frequency of Semantic Cache Module (SCM) calls, which indicates when the LLM fails to provide timely guidance.

### C.3.2 Results and Analysis

Table 3 presents the experimental results across all three scenarios and prompt variations.

Table 3: System Performance Across Different Prompt Formulations (Medium Traffic Density)

| Driving Conditions | Prompt Sets | SR (%) | CR (%) | SCM Calls (%) |
|---|---|---|---|---|
| Overtaking | Prompt A (Detailed) | 91.0 | 9.0 | 19.0 |
| | Prompt B (Concise) | 88.0 | 12.0 | 4.0 |
| | Prompt C (Original) | 93.0 | 7.0 | 4.0 |
| Merging | Prompt A (Detailed) | 92.0 | 8.0 | 21.0 |
| | Prompt B (Concise) | 90.0 | 10.0 | 4.0 |
| | Prompt C (Original) | 92.0 | 8.0 | 6.0 |
| Trilemma | Prompt A (Detailed) | 62.0 | 38.0 | 27.0 |
| | Prompt B (Concise) | 61.0 | 39.0 | 5.0 |
| | Prompt C (Original) | 69.0 | 31.0 | 7.0 |

**Overall Robustness**: The experimental results indicate that the HCRMP framework exhibits a degree of robustness to prompts with different styles. While performance varied across prompts—most notably in the challenging "Trilemma" scenario—the framework maintained reasonable safety performance across all tests, with SR ranging from 61% to 93% and CR from 7% to 39%.

**Counterintuitive Effect of Detailed Prompts**: Notably, the detailed prompt (Prompt A), despite providing more information, failed to improve and in some cases even degraded performance compared to the original prompt (Prompt C). For instance, in the Trilemma scenario, Prompt A achieved only 62% SR compared to 69% for Prompt C. Simultaneously, it dramatically increased the frequency of Semantic Cache Module (SCM) calls—reaching 27% in Trilemma compared to just 7% for Prompt C. This indicates that overly complex prompts may introduce counterproductive noise or constraints, leading to a clear trade-off where both safety performance and timeliness are negatively impacted.

**Balance Between Conciseness and Detail**: The concise prompt (Prompt B) exhibited moderate performance, generally falling between the detailed and original versions. While it reduced SCM call frequency compared to Prompt A, it did not achieve the optimal balance of safety and efficiency provided by Prompt C.

**Implications for Prompt Design**: These findings suggest that prompt engineering for safety-critical systems requires careful consideration. Excessive detail can overload the LLM's reasoning process and increase inference latency (reflected in higher SCM calls), while overly concise prompts may lack sufficient context for optimal decision-making. The original prompt (Prompt C) achieves the best overall effectiveness by ensuring lower inference latency without sacrificing core safety metrics, making it the optimal choice within the HCRMP framework.

**System Adaptability**: Importantly, even with suboptimal prompts, the system maintained functional performance due to the LLM-Hinted paradigm's inherent safety mechanisms: (1) the RL agent retains self-optimization capability to buffer poor LLM guidance, and (2) the Semantic Cache Module provides fallback when LLM outputs are delayed, ensuring continuous operation.

### C.3.3 Prompt Versions

Below we present the three prompt versions used in the robustness evaluation. These prompts are designed for the Contextual Stability Anchor (CSA) module to generate multi-critic weights.

---

**Prompt A: Detailed Version**

**Objective:** You are an expert autonomous driving system analyst. Your task is to carefully analyze the current driving scenario and generate a three-element weight vector that represents the relative importance of three critical driving attributes: Safety, Comfort, and Efficiency.

**Input Information:**
- Ego vehicle position: (`{ego_position_x}`, `{ego_position_y}`)
- Ego vehicle velocity: (`{ego_velocity_x}`, `{ego_velocity_y}`)
- Surrounding environment: `{surrounding_info}`
- Scenario type: `{scene_flag}`
- Reference knowledge: `{rag_output}`

**Detailed Instructions:**
Please perform a comprehensive analysis of the driving scenario considering traffic density, road conditions, nearby obstacles, and potential risks. Based on this analysis and the provided reference knowledge from traffic regulations, determine how to balance the three driving attributes.

The three elements of the output vector represent:
1. **Safety**: Prioritize collision avoidance, maintaining safe distances, and responding to hazards.
2. **Comfort**: Ensure smooth acceleration/deceleration and gentle steering to enhance passenger experience.
3. **Efficiency**: Optimize travel time and maintain appropriate speed while respecting traffic flow.

**Requirements:**
- The sum of the three elements must equal 1.0
- Each element must be a positive decimal value between 0 and 1
- Round each value to 2 decimal places
- In high-risk scenarios (e.g., dense traffic, pedestrians nearby), prioritize Safety
- In low-risk scenarios, balance all three attributes more evenly

**Output Format:** Provide only the three-element vector in the format: [Safety, Comfort, Efficiency]
Example: [0.65, 0.20, 0.15]

---

**Prompt B: Concise Version**

Generate a weight vector for autonomous driving attributes.
**Input:**
- Ego position: (`{ego_position_x}`, `{ego_position_y}`)
- Ego velocity: (`{ego_velocity_x}`, `{ego_velocity_y}`)
- Environment: `{surrounding_info}`

**Output:** Three-element vector [Safety, Comfort, Efficiency] where elements sum to 1.0.
Requirements: 2 decimal places, prioritize safety in high-risk scenarios.

# D  Experiments and Results

## D.1  Computational Resources

The model training experiments in this study are accomplished relying on the following computational resources: a total of four NVIDIA A100-PCIE GPUs (40GB of graphics memory on a single card) are used. Each GPU is configured with 10 virtual CPU cores (Intel Xeon Gold 6248R) and 72GB of system memory. For each model, 200 epochs are performed, with each epoch consisting of 5 episodes.

## D.2  Experiments setup

Town02 provides a variety of challenging driving conditions for evaluating basic driving ability and complex decision-making actions, as it encompasses circular road networks, regular neighborhoods, and multi-level interchanges. We introduce varying traffic density settings as follows: in the on-ramp merging scenario, low, medium, and high densities correspond to 2, 4, and 8 surrounding vehicles, respectively, located on the two lanes adjacent to the merging lane on the main road. In the multi-lane overtaking scenario, these levels represent 1, 2, or 3 surrounding vehicles ahead of the ego vehicle.

For safety-critical trilemma and occluded pedestrian scenarios, the definitions for low, medium, and high traffic flow densities are unified. These are determined by counting the number of surrounding vehicles in the "borrowed lane" (i.e., the adjacent/oncoming lane that the autonomous vehicle temporarily occupies or interacts with for specific maneuvers). Specifically, low, medium, and high densities correspond to the presence of 2, 3, and 4 surrounding vehicles, respectively, in the aforementioned "borrowed lane".

These traffic density levels are designed to create diverse traffic environments and interaction situations, thereby increasing task complexity and enabling a more comprehensive evaluation of the algorithm's decision-making ability and adaptability under different traffic conditions.

## D.3  HCRMP with Varied LLM

**Our study primarily conducts experimental validation based on the Gemini-2.5-Pro model.** To comprehensively assess the generalization capability and performance of the HCRMP architecture across different LLMs, we further evaluated its effectiveness using four mainstream LLMs: GPT-4o, Llama-3.3-70B-Instruct, Deepseek-r1, and Qwen-Turbo in trilemma's high-density driving condition.

Experimental results show that, as shown in figure 6 and table 4, the overall performance of HCRMP remains consistent across different LLM. Although these LLM exhibit some variation in hallucination-free rate evaluations, their differences have minimal impact when serving as semantic prompt sources for HCRMP. This demonstrates that HCRMP, as an LLM-Hinted RL paradigm, can effectively

leverage the semantic prompting capabilities of various LLM. Through the self-learning mechanism of the RL agent, it compensates for disparities in LLM performance, thereby achieving similarly strong driving performance regardless of the underlying LLM backend.

Table 4: HCRMP with Varied LLM

| | SR(%) | CR(%) | AS(m/s) | TD(m) | TS(s) | SV(m/s) | AV(m/s$^2$) |
|---|---|---|---|---|---|---|---|
| HCRMP-Gemini-2.5-Pro | 64 | 36 | 9.94 | 77.34 | 47.13 | 9.96 | 1.72 |
| HCRMP-GPT-4o | 61 | 39 | 6.85 | 53.09 | 52.88 | 2.40 | 1.84 |
| HCRMP-Deepseek-r1 | 61 | 39 | 8.26 | 87.35 | 51.44 | 8.46 | 1.97 |
| HCRMP-Qwen-Turbo | 59 | 41 | 7.02 | 65.04 | 56.84 | 2.17 | 1.79 |
| HCRMP-Llama-3.3-70B-Instruct | 63 | 37 | 6.84 | 47.68 | 51.70 | 2.26 | 1.79 |

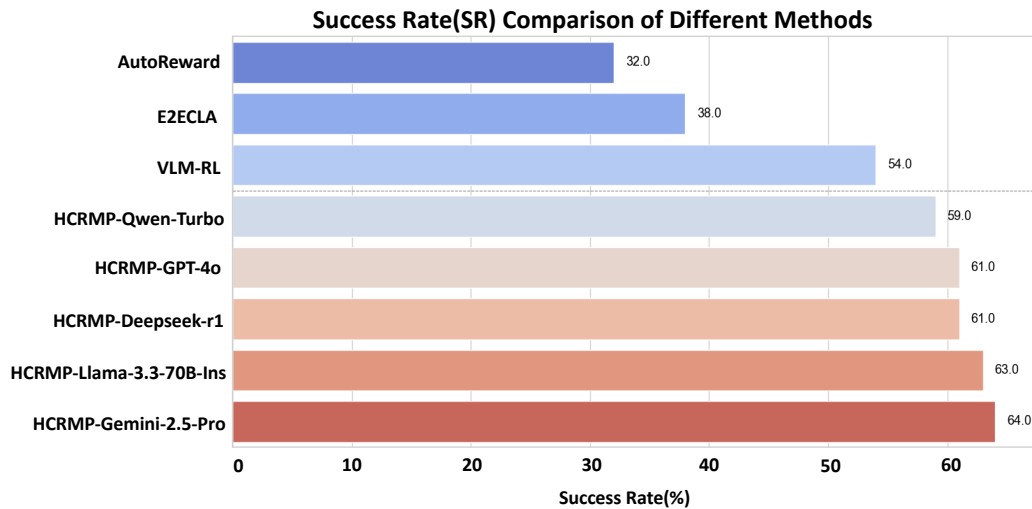

Figure 6: Success Rate (SR) Comparison of Different Methods

## D.4 Impact of LLM Hallucinations on Q-value Estimation

A critical challenge for LLM-integrated RL systems is understanding how incorrect LLM outputs lead to incorrect Q-value estimates or rewards. To address this, we conducted a controlled experiment to establish a quantitative link from an erroneous LLM prompt, to a distorted Q-value estimation, and ultimately to degraded driving performance.

### D.4.1 Experimental Setup

#### D.4.1.1 Scenario
The experiment was conducted in a high-density overtaking scenario (as described in Section 4.1 of the main paper), which requires careful judgment of when it is safe to accelerate and overtake surrounding vehicles.

#### D.4.1.2 Control and Experimental Groups

- **Control Group**: Instances where HCRMP receives correct LLM outputs. For example, the LLM correctly identifies a construction site ahead and outputs appropriate semantic hints (e.g., elevated safety weight, recognition of hazard in state representation).

- **Experimental Group**: Instances specifically selected where the LLM produces hallucinated outputs. For example, the LLM fails to recognize a construction site ahead, providing incorrect semantic hints that do not reflect the true risk level.

### D.4.1.3 Measurement

We focus on the Q-value estimation for the critical action "accelerate to overtake" ($a_{\text{overtake}}$) and measure:

- **Q-value Estimation Error**: The relative difference between Q-values estimated under correct vs. incorrect LLM outputs, calculated as:

$$\text{Q-value Error} = \frac{|Q_{\text{incorrect}}(s, a_{\text{overtake}}) - Q_{\text{correct}}(s, a_{\text{overtake}})|}{|Q_{\text{correct}}(s, a_{\text{overtake}})|}$$

  where $Q_{\text{correct}}$ is the Q-value estimated with correct LLM output, and $Q_{\text{incorrect}}$ is the Q-value with hallucinated LLM output.

- **Task Success Rate**: The percentage of successful task completions (collision-free overtaking) in each group.

### D.4.2 Results

Table 5 presents the quantitative impact of LLM hallucinations on Q-value estimation and driving performance.

Table 5: Impact of LLM Hallucinations on Q-value Estimation and Task Performance

| Method | Estimated Q-value Error (%) | Success Rate (%) |
| --- | --- | --- |
| HCRMP (Correct LLM Output) | – | 85.0 |
| HCRMP (Incorrect LLM Output) | 23.0 | 67.0 |
| Performance Degradation | +23.0% error | -18.0% SR |

### D.4.3 Analysis and Implications

#### D.4.3.1 Quantitative Evidence of Hallucination Impact

The experimental results provide concrete empirical numbers demonstrating how LLM hallucinations propagate through the system:

- **Q-value Distortion**: When the LLM produces hallucinated outputs (e.g., failing to recognize a construction hazard), the estimated Q-value for the "accelerate to overtake" action exhibits a 23% error compared to the correct estimation. This distortion occurs because:
  - The incorrect semantic state representation from ASR module fails to capture the true risk level.
  - The CSA module, misled by the incorrect scenario understanding, assigns inappropriate critic weights (e.g., underweighting safety).
  - The resulting integrated advantage function $\hat{A}_{\text{int}}(s, a)$ (Eq. 2 in main paper) becomes biased, leading to overoptimistic Q-value estimates for dangerous actions.

- **Performance Degradation**: The 23% Q-value estimation error directly translates to an 18 percentage point drop in task success rate (from 85% to 67%). This demonstrates the causal chain: *hallucinated prompt → distorted Q-value → dangerous behavior (collision)*.

- **Why HCRMP Still Maintains 67% Success**: Importantly, even with hallucinated LLM outputs, HCRMP maintains a 67% success rate rather than complete failure. This robustness is attributed to the LLM-Hinted paradigm's design:
  - The RL agent retains self-optimization capability based on environmental feedback, allowing it to partially correct for LLM errors through policy learning.
  - The Semantic Cache Module (SCM) can replace extremely poor LLM outputs with more reliable cached weights from similar historical scenarios.
  - The weakly coupled design prevents LLM hallucinations from completely dominating the decision-making process.

# E    Limitations and Broader Impacts

While HCRMP demonstrates promising motion planning capabilities, its current limitations primarily involve potential inter-module information loss and the sim-to-real gap inherent in simulator-based evaluations, defining clear avenues for future work. Future work will enhance data fusion and module integration to reduce information loss, and validate HCRMP in real-world tests. This will strengthen HCRMP, an LLM-hinted RL paradigm that promises to significantly advance intelligent transportation.

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
