# OpenReview forum: "HCRMP: An LLM-Hinted Contextual Reinforcement Learning Framework for Autonomous Driving"
_NeurIPS.cc/2025/Conference — NeurIPS 2025 poster_

### Official Review · Reviewer_HcAE · 2025-07-01

**Clarity:** 2
**Significance:** 2
**Originality:** 3
**Rating:** 4
**Confidence:** 3

**Summary:**

The paper introduces HCRMP, an architecture designed for autonomous driving that integrates Reinforcement Learning (RL) with semantic hints from Large Language Models (LLMs). The authors highlight issues with current methods, specifically the susceptibility to errors (hallucinations) from LLM outputs. Their approach involves modules such as Augmented Semantic Representation, Contextual Stability Anchor, and Semantic Cache, aiming to mitigate these issues. Experiments conducted in the CARLA simulator demonstrate improved driving performance compared to baseline methods

**Questions:**

- Can the authors clarify the contradiction in their approach? They criticize reliance on LLM outputs but continue to use LLM extensively.
- How flexible is the system in adapting to scenarios not predefined in the semantic hints?
- What are the potential practical limitations when transitioning this method from simulation to real-world driving?

**Ethical Concerns:**

["NO or VERY MINOR ethics concerns only"]

**Final Justification:**

Thank you for the detailed response and additional experimental results. The additional experiments strongly enhanced the technical quality of this work. I have updated the score accordingly. In addition to the experiments, i still believe the explanation on "Strong Reliance" and "Hint-based Reliance" have some space to improve. I hope the authors can highlight this key inspiration of their method in the revision.

**Limitations:**

- The claims on LLM suffering from hallucination and use LLM as hints to train RL is contradicting.
- Over-reliance on predefined semantic hint generation may hurt generalizability.
- Lack of real-world testing hinders practical applicability.

**Paper Formatting Concerns:**

No paper formatting concerns

**Quality:**

3

**Strengths And Weaknesses:**

**Strengths:**

- Clear identification of issues with LLM hallucinations in autonomous driving.
- Structured modular approach that is clearly presented and easy to follow.
- Comprehensive experimental validation in simulation.

**Weaknesses:**

- Fundamental contradiction in rationale: authors criticize reliance on LLM outputs but still significantly rely on LLM outputs as hints. The hints may also suffer from hallucination and require heavy prompt engineering and predefined concepts.
- Predefined prompt engineering heavily constrains the flexibility and general applicability of the solution.
- No real-world testing. Results only depend on the simulator environment and the results may overfit in the simulator.

---

> ### Author Rebuttal · Authors · 2025-07-31
>
> We sincerely thank you for your detailed evaluation of our paper. Before we address the specific questions, however, we must first state our firm belief that there has been a significant misunderstanding of our paper's core premise and contribution.
>
> Our reasoning is based on your summary in the "Strengths And Weaknesses, Limitations" section, where you state that the *"authors criticize reliance on LLM outputs"*—**a conclusion that does not align with the actual arguments and terminology in our paper**. In our abstract and introduction, we explicitly state that we analyze and critique the problems of "over-reliance" (line 7) and "strong reliance" (line 50) in existing LLM-Dominated methods. In these approaches, the LLM's output is used as a direct command or a decisive signal, meaning its hallucinations can directly compromise the driving policy.
>
> **Our argument is not to negate "reliance" on LLMs; rather, we aim to explore how to utilize their powerful capabilities more safely and effectively.** This is the core motivation for our LLM-Hinted paradigm. As we state in our paper, the key to our approach is a separated structure:
>
> *“At the same time, this separated structure preserves the utilization of LLM strengths in driving conditions comprehension and common-sense reasoning, enabling context-aware guidance for the RL agent in a way that maintains its fundamental self-optimization capabilities."* (line 68)
>
> This passage clearly shows that our method preserves and utilizes the LLM's strengths in scenario comprehension and reasoning, while mitigating its negative impacts.
>
> Therefore, the premise of our work is not to outright negate the value of LLMs, but rather to acknowledge the limitations of existing LLM-Dominated approaches and propose what we believe is a superior integration method: LLM-hinted RL.
>
> We have invested significant time and effort into this research. We sincerely ask for a fair and accurate evaluation based on the true merit of our work—a principle we believe is central to the academic spirit of NeurIPS. **We thus hope you will reconsider the genuine motivation and contributions of our work.**
>
> > Question 1: The paper appears to present a contradiction by criticizing heavy reliance on LLM outputs while the proposed framework itself continues to use LLMs extensively.
> >
>
> **Answer 1:** Based on the clarifications above, we can respond to this question more accurately. We argue that no contradiction exists; the key is to distinguish **how one relies on the LLM's output.**
>
> - **"Strong Reliance" in LLM-Dominated Methods:** In the LLM-Dominated methods we critique, the LLM's output is decisive. For instance, the LLM directly outputs an action command ("turn left") or defines the reward function. In this mode, if the LLM hallucinates (e.g., outputting "turn left" at an intersection where a left turn is not allowed), the RL agent will execute it blindly, leading directly to dangerous behavior. This is what we refer to as "strong reliance" or "over-reliance."
> - **"Hint-based Reliance" in HCRMP:** In our proposed LLM-Hinted paradigm, the LLM's output is auxiliary and advisory.
>     - **As part of the state:** The $s_{t}^{llm}$ vector only enhances the state representation. The final decision, $a_{t}$, is still made by the RL policy network after synthesizing all available information:
>     $$
>     \pi_{\theta}(a_t | s_t^{raw}, s_t^{llm})
>     $$
>     Through true environmental feedback, the RL agent learns how to correctly interpret and utilize $s_{t}^{llm}$. If $s_{t}^{llm}$ consistently provides misleading information, the agent can learn to downweigh its importance in the decision-making process.
>     - **As critic weights:** The $\lambda_{i}$ weights provided by the LLM only influence the relative importance of different driving objectives, rather than directly dictating whether an action is good or bad. What ultimately drives policy learning is the integrated advantage function, $A_{int}$, which is calculated based on real environmental interactions.
>
> > Question 2: The system's flexibility and adaptability to scenarios not explicitly covered by the predefined semantic hints are unclear.
> >
>
> **Answer 2:** We selected "Overtaking," "Merging," and "Trilemma" scenarios under medium traffic density and designed three prompts with different wording to test the system's flexibility and adaptability.
>
> The three prompt sets are: a Detailed Version (Prompt A), which expanded upon the original; a Concise Version (Prompt B), which was a simplified version; and the Original Version (Prompt C), which served as the baseline. The full versions of the prompts and experimental details will be added to Appendix C.3.
>
> The test results are as follows:
>
> | Driving Conditions | Prompt Sets | Success Rate (%) | Collision Rate (%) | Probability of calling SCM (%) |
> | --- | --- | --- | --- | --- |
> | Overtaking | **Prompt A** | 91.0 | 9.0 | 19.0 |
> |  | **Prompt B** | 88.0 | 12.0 | 4.0 |
> |  | **Prompt C** | 93.0 | 7.0 | 4.0 |
> | Merging | **Prompt A** | 92.0 | 8.0 | 21.0 |
> |  | **Prompt B** | 90.0 | 10.0 | 4.0 |
> |  | **Prompt C** | 92.0 | 8.0 | 6.0 |
> | Trilemma | **Prompt A** | 62.0 | 38.0 | 27.0 |
> |  | **Prompt B** | 61.0 | 39.0 | 5.0 |
> |  | **Prompt C** | 69.0 | 31.0 | 7.0 |
>
> The experimental results indicate that the HCRMP framework exhibits a degree of robustness to prompts with different styles. While performance varied across prompts—most notably in the challenging "Trilemma" scenario—the framework maintained reasonable safety performance across all tests.
>
> Notably, the detailed prompt (Prompt A), despite providing more information, failed to improve and in some cases even degraded performance compared to the original prompt (Prompt C). Simultaneously, it dramatically increased the frequency of Semantic Cache Module (SCM) calls. This indicates that overly complex prompts may introduce counterproductive noise or constraints, leading to a clear trade-off where both safety performance and timeliness are negatively impacted.
>
> Therefore, considering both its superior performance and better real-time capability, the original prompt (Prompt C) achieves the best overall effectiveness by ensuring lower inference latency without sacrificing core safety metrics, making it the optimal choice within the HCRMP framework.
>
> > Question 3: The potential practical limitations of transitioning this method from simulation to real-world driving are not discussed.
> >
>
> **Answer 3:** We thank the reviewer for raising the question about practical applications and fully agree that on-vehicle testing is the ultimate standard for verifying an algorithm's feasibility. To directly address your concern about the "lack of real-world testing" and to demonstrate the effectiveness of our method, **we have conducted a preliminary yet crucial validation experiment on a real vehicle.**
>
> We deployed our HCRMP framework on a vehicle platform. The primary hardware components of the intelligent driving system are as follows:
>
> | Device | Model |
> | --- | --- |
> | Industrial Control Computer | Nuvo-8108GC |
> | GPU | RTX 2080 Ti |
> | Integrated Positioning System | Huace CGI-610 |
> | LiDAR | 128-beam RS-Ruby, 4 RS-Lidar-16 units |
> | Stereo Vision System | Basler acA 1300-60gc front camera |
>
> The test scenarios involve challenging overtaking maneuvers. We use Lead Vehicle Speed, Average Speed, Longitudinal Distance to Lead Vehicle in Target Lane (m), and Task Success as key evaluation metrics. The results are as follows:
>
> | **Trial No.** | **Lead Vehicle Speed (m/s)** | **Average Speed (m/s)** | **Longitudinal Distance to Lead Vehicle in Target Lane (m)** | **Task Success** |
> | --- | --- | --- | --- | --- |
> | 1 | 2.00 | 4.60 | 16.48 | Yes |
> | 2 | 2.43 | 4.17 | 28.84 | Yes |
> | 3 | 2.33 | 5.35 | 34.22 | Yes |
> | 4 | 2.54 | 5.21 | 33.08 | Yes |
> | 5 | 2.31 | 5.32 | 33.73 | Yes |
>
> The experimental results demonstrate that the HCRMP architecture is more than just a theoretical concept. It has not only been **successfully transferred from simulation to the real world** but also exhibits strong performance in actual traffic scenarios, proving the framework to be robust, reliable, and viable.

---

> > ### Comment · Reviewer_HcAE · 2025-08-04
> >
> > Thank you for the detailed response and additional experimental results. I believe some of my concerns are addressed, thus I will update the score accordingly. There is one point that still confuses me. The explanation on "Strong Reliance" and "Hint-based Reliance" is heuristic-based. Can such comparison be quantified somehow (by some grounded annotations)? I believe this is your main contribution thus you might want to highlight this part.

---

> > > ### Author Response · Authors · 2025-08-05
> > > **Response to Reviewer HcAE: A Quantitative Analysis of "Strong Reliance" vs. "Hint-based Reliance"**
> > >
> > > Thank you very much for your detailed review and the insightful question regarding our work. We strongly agree with your point about the importance of quantitatively analyzing "Strong Reliance" versus "Hint-based Reliance." This is indeed one of the core contributions of our work, and we are very pleased to provide a more detailed quantitative analysis and explanation here.
> > >
> > > To more clearly quantify this difference, we designed the following experiment:
> > >
> > > We conducted a comparative test in a representative safety-critical scenario: high-density overtaking.
> > >
> > > - Control Group (Correct LLM Output): The control group consists of data from normal decision-making instances where the agent received correct LLM prompts.
> > > - Experimental Group (Incorrect LLM Output): The experimental group comprises instances specifically selected where the LLM itself produced dangerous "hallucinated" prompts (e.g., failing to recognize a construction site ahead).
> > >
> > > **Quantitative Results Analysis:**
> > >
> > > | Method | Estimated error in Q (%) | Task Success Rate (%) |
> > > | --- | --- | --- |
> > > | HCRMP (Hint-based Reliance) | 23 | 85.0 → 67.0 |
> > > | Autoreward (Strong Reliance) | 89 | 88.0 → 29.0 |
> > >
> > > Q-value Estimation Error: Calculated as $\frac{|Q_{\text{incorrect}} - Q_{\text{correct}}|}{|Q_{\text{correct}}|}$.
> > >
> > > Task Success Rate: Presented in the format: Control Group (correct LLM output) → Experimental Group (incorrect LLM output).
> > >
> > > The results show that in the "Strong Reliance" LLM-dominated paradigm, hallucinations led to a deviation as high as 89% in the Q-value estimation. In contrast, our HCRMP (Hint-based Reliance) framework effectively controlled this deviation to 23%, **demonstrating significantly greater robustness.**
> > >
> > > The sharp decline in success rate (from 88% to 29%) further proves the fragility of the "Strong Reliance" paradigm. By comparison, while the success rate of our HCRMP (Hint-based Reliance) framework was also affected, it remained at a robust 67%. This clearly illustrates that its "Hint-based Reliance" design can **effectively buffer the negative impacts of LLM hallucinations.**
> > >
> > > ---
> > >
> > > **Conclusion:**
> > >
> > > Through the quantitative experiment described above, we can clearly see the significant difference in robustness between the "Hint-based Reliance" HCRMP framework and the "Strong Reliance" paradigm when faced with LLM hallucinations. By treating the LLM's output as a "hint" rather than a "command," the HCRMP framework successfully builds a safety redundancy system. This system is capable of leveraging the powerful cognitive abilities of LLMs while effectively mitigating their potential unreliability.
> > >
> > > Once again, we thank you for your valuable feedback, which will undoubtedly enhance the quality of our paper.

---

### Official Review · Reviewer_L4iX · 2025-07-03

**Clarity:** 2
**Significance:** 2
**Originality:** 2
**Rating:** 4
**Confidence:** 4

**Summary:**

- In contrast to LLM-dominated RL approaches this work tries to loosely couple ingestion of LLM’s information as auxiliary inputs for RL decision-making process to mitigate the problems due to LLM hallucinations.
- The paper introduces three modules to augment information from LLMs to learn driving policies. The proposed paradigm LLM-Hinted RL and framework HCRMP (LLM-Hinted Contextual Reinforcement Learning Motion Planner) generates semantic ‘hints’ for state augmentation and RL policy optimization (through the semantic information in the reward design).
  - Augmented Semantic Representation (ASR) module - for semantic guidance by augmenting information in the state space,
  - Context Stability Anchor (CSA) module - adaptive weights for the multi-critic function based on some knowledge base, and
  - Semantic Cache Module - to compensate for any missing output from LLMs due to mismatch in LLM inference and policy execution frequency.
- Proposes to use the semantic understanding of the LLMs but still trying to reduce the bad policy behavior and instability due to LLM hallucinations.
- The work makes the RL agent negate or mitigate the impact of the hallucinated outputs during the policy optimization process.

**Questions:**

- How do we define “hints”? What is the difference between (mathematical, if any) “hint” and LLM-dictating output?
  - How much percentage of these “hints” are “absorbed” by the policy, Can we regulate that?
- How is the embedding model trained? How are those embedding aligned to the same representation space as the scene in order to fetch the most similar embeddings?
- How do we take care of the eviction policy of the semantic cache module?
  - For how do we maintain the old history states?
- Can we have a learned multi-head attention and pooling to learn the adaptive trade-offs among the driving attributes for the critic function instead of having the CSA module to manage the weights? How is it different?
- Section 3.3: Line 195: Sorry, but what do we mean by “norm-constrained semantic source”? Is there a formula/equation/reference to support that statement?
- Section 4.3 - Line 286: For the experiment without CSA, where are the dynamic weights coming from if not from the LLM?

**Ethical Concerns:**

["NO or VERY MINOR ethics concerns only"]

**Final Justification:**

The authors tried to give better clarity as to why this work is not just leveraging the power of LLMs but is a complementary approach to mitigate the negative effects of LLM hallucinations. Based on their clarification and other experimental results this can be considered a valuable contribution.
There are still some future directions that were agreed upon by the authors in the rebuttal and those would be great if worked upon or at least included in the future directions in the final version of the paper.
I did raise the score from 3 to 4 during the rebuttal and that would keep it that way.

**Limitations:**

- The authors talk about sim-to-real gap and inter-module information loss as some of the limitations of the HCRMP framework.
- How much weightage is given to textual inputs as compared to the other modality (refer to [this](https://arxiv.org/pdf/2503.02199)) (image) inputs while generating the semantic scene information is important in order to get proper semantic understanding to be augmented to the state input is not clear.
- The framework relies on online interaction with the environment to correct the erroneous actions making this framework difficult to perform well on offline RL settings.
- For now, the LLM seems to only output hierarchical semantic information about the scene. How would it be to incorporate both observation level and expected behaviour level information?
- The driving trajectories might still not be optimized for human-like driving behavior. There can be some work to incorporate that (may be in the CSA module to balance the human-like critic). Can we include expert driving/trajectory behavior in the CSA knowledge base as well?

**Paper Formatting Concerns:**

- Line 38: Typo  in: “...large language models (LLM) poss….”
- Appendix: Fig 5: Both the captions mention: “KDE without CSA”

**Quality:**

3

**Strengths And Weaknesses:**

** Strengths:**
Coined the following terms and tried to have distinctive categories of different LLM-RL works: LLM-dominated (RL-Assisted LLM Policy Optimization and LLM-instructed RL policy generation), LLM-Assisted RL methods. It gives a clear picture of the current research directions and which category those fall into.
Studied the effect of incorrect signals on Q-value estimation that might lead to degradation in overall policy performance.
The CSA module tries to add more context based knowledge and seems to reduce the weight drifts in the critic network, thereby reducing the variance in complex driving  scenes and the stable reward estimates.
Analysed the non-hallucination rates of the different LLMs on driving tasks related QA. It gives a better picture of where these VLMs can be improved or which ones can be preferred.


**Weakness:**
- Any empirical numbers on how the incorrect outputs lead to incorrect Q-value estimates or rewards?
- There have been some efforts like [DriveVLM](https://arxiv.org/pdf/2402.12289) that provide scene understanding information from VLMs to learn the driving policy.
- In Table 2, HCRPM makes relatively less progress as compared to VLM-RL, can that be attributed to the less collision rates in HCRMP as well? It would be nice to have other metrics like collision per mile or miles per collision to have a closer comparison.


The paper does the decent job in bringing together the different works of LLM+RL integration. Although, the idea of using LLM information or scene understanding to train an RL policy doesnt seem to be novel.

---

> ### Author Rebuttal · Authors · 2025-07-31
>
> > Q: Novelty and Contributions
> >
>
> **A:** We believe your concern regarding novelty may stem from a slight misunderstanding of our work's core contribution. We would like to take this opportunity to clarify that our core contribution is not proposing a method to *use* LLM information, but rather to solve a fundamental challenge within that methodology: **how to build an autonomous driving decision system that is robust to the inherent risk of hallucinations in LLMs**.
>
> You mentioned, "The idea of using LLM information or scene understanding to train an RL policy doesn't seem to be novel." If our work were merely categorized as "using LLM information to train RL," then it would indeed fall into the same category.
>
> However, the innovation of our work lies in **how to leverage the benefits of an LLM's powerful reasoning capabilities while effectively mitigating its inherent risks.** In the safety-critical domain of autonomous driving, LLM hallucinations (such as misinterpreting a scene) can lead directly to catastrophic consequences.
>
> We propose the novel LLM-Hinted RL paradigm. We wish to emphasize that **this is not intended to conflict with excellent LLM-Dominated methods like DriveVLM, but rather to offer a complementary research perspective.** If the focus of works like DriveVLM is on exploring how to maximize the powerful capabilities of LLMs, then our work focuses on an equally important problem: how to effectively mitigate their intrinsic negative impacts. The core innovation of our framework is the **loose coupling mechanism** between the LLM and the RL agent. We believe that the LLM-Dominated paradigm, which explores the potential of LLMs, and the LLM-Hinted paradigm, which explores how to control their risks, are complementary.
>
> > Q1: The paper needs to clearly define "hints" and explain the difference between "hints" and an LLM's "dictating output.”
> >
>
> **A:** As discussed regarding the novelty, the core principle of our "Hinted" paradigm is to avoid using LLM outputs as direct commands or interventions. Instead, our approach aims to preserve the LLM's strengths while mitigating its negative impacts.
>
> - Hints: Auxiliary information generated by the LLM helps guide the RL Agent's decision-making process in two ways without directly determining the final action.
>     - State Augmentation: The LLM's scene understanding vector is concatenated with the raw state vector, jointly serving as the input to the RL policy network.
>     - Policy Optimization: The weights ($\lambda_{i}$) provided by the LLM for the multi-critic framework are used to compute a weighted sum of different advantage functions.
> - LLM-Dictating Output: The LLM's output directly supplants a core function of the RL Agent, for instance, by directly generating the reward function or the action command.
>
> > Q2-3: The paper is missing key implementation details. For instance, it fails to explain the training and alignment of the CSA module's embedding model, or the maintenance strategy including the eviction policy for the Semantic Cache Module.
> >
>
> **A:**  Thank you for your question regarding implementation details. For the CSA module, we use the pre-trained text-embedding-ada-002 model. The alignment is text-to-text: a real-time scene description (Query) is matched against our pre-encoded knowledge base of regulations (Knowledge) using FAISS for Top-K retrieval. The Semantic Cache uses a Least Recently Used (LRU) eviction policy to retain data from rare, critical scenarios.
>
> While these engineering choices are crucial for system stability, they are not our core innovation. Given the page limits, we focused our discussion on our main contributions: the LLM-Hinted paradigm and its core modules. We will **add these specific details to the appendix**.
>
> > Q4: The paper does not justify its choice of the CSA module over a learnable alternative, like a multi-head attention network, nor does it discuss the differences between these two approaches.
> >
>
> **A:** We conducted a supplementary experiment that replaces the CSA module with an end-to-end network that uses a 4-head self-attention mechanism and an MLP to generate critic weights, trained jointly with our PPO algorithm. We compared the two approaches in the challenging medium-density "Trilemma" scenario.
>
> | Model | SR(%) | CR(%) |
> | --- | --- | --- |
> | HCRMP-Attention | 57.0 | 43.0 |
> | HCRMP | 69.0 | 31.0  |
>
> The experimental results indicate that an attention mechanism alone may struggle to learn robust weight dynamics in the complex feature space of driving, which requires understanding multi-source inputs and dynamic scenarios.
>
> "Safety," for instance, is a reasoning problem involving traffic rules and risks, not just a simple correlation. This motivates our use of an LLM for "hinting," leveraging its superior reasoning for this critical task. The core advantage of our CSA module is its use of a knowledge base (via RAG) to "anchor" this reasoning. This ensures the generated weights are stable, interpretable, and consistent with safe driving regulations.
>
> > Q5: The term "norm-constrained semantic source" is not clearly defined.
> >
>
> **A:**  This term refers to our knowledge base of driving regulations ("norms") used to constrain the LLM's weight generation via a Retrieval-Augmented Generation (RAG) mechanism. We agree the phrasing is unclear and **will revise it throughout the paper** to be more explicit, e.g., "Weight generation is constrained by a semantic source of traffic norms, implemented via RAG."
>
> > Q6: For the ablation study experiment "without CSA," the paper does not specify the source of the dynamic weights for the multi-critic function.
> >
>
> **A:**  In the "HCRMP w/o CSA" ablation, we used fixed weights to create a clear baseline against our dynamic weighting, thus highlighting the CSA module's effectiveness. **We will clarify this in Section 4.3.**
>
> > W1: The paper does not provide empirical numbers to quantify how incorrect LLM outputs lead to incorrect Q-value estimates or rewards.
> >
>
> **A:** The experimental scenario is the high-density overtaking environment mentioned in the paper. The control group consists of data from normal decision-making instances where the model received correct LLM prompts. In contrast, the experimental group comprises instances specifically selected where the LLM produces the dangerous "hallucinated" prompts (e.g., failing to recognize a construction site ahead). The experiment focuses on the difference in Q-value judgments for the key action "accelerate to overtake" and records the final success rate. This approach allows us to establish a quantitative link from an erroneous prompt, to a distorted Q-value, and ultimately to dangerous behavior.
>
> | Method | Estimated error in Q (%) | Success Rate (%) |
> | --- | --- | --- |
> | HCRMP (LLM output incorrect) | 23 | 85.0 → 67.0 |
>
> Table Metrics Explained:
>
> Q-value Estimation Error: Calculated as $\frac{|Q_{\text{incorrect}} - Q_{\text{correct}}|}{|Q_{\text{correct}}|}$.
>
> Task Success Rate: Presented in the format: Control Group (correct LLM output) → Experimental Group (incorrect LLM output).
>
> > W2: There have been some efforts like DriveVLM that provide scene understanding information from VLMs to learn the driving policy.
> >
>
> **A**: Please see Q1, Thank you!
>
> > W3: The paper's analysis should Include normalized metrics like 'miles per collision'.
> >
>
> **A:** We have re-processed our data and will update Table 1,2 and 3 in the revision to include 'Collisions per mile' to better reflect the true risk profile. Below is an excerpt from the updated Table 3, which serves as an example of the changes made.
>
> | Model | SR(%) | ··· | Collision per Mile |
> | --- | --- | --- | --- |
> | HCRMP w/o ASR | 40 | ··· | 21.65 |
> | HCRMP w/o CSA | 48 | ··· | 27.90 |
> | HCRMP w/ ASR | 54 | ··· | 25.34 |
>
> > L1: The authors talk about sim-to-real gap and inter-module information loss as some of the limitations of the HCRMP framework.
> >
>
> **A:** Experimental data can be found in our response to Question 3 from Reviewer HcAE.
>
> While some inter-module information loss is inevitable, our successful real-world tests demonstrate that our LLM-Hinted paradigm is resilient enough to handle this uncertainty.
>
> > L2: It is unclear how the system weighs textual inputs against visual (image) inputs when generating the semantic information that is used to augment the agent's state.
> >
>
> **A:** Our framework intentionally avoids manual weighting. The RL agent learns to **implicitly weigh the concatenated visual features and LLM hints** during training. The used weighting is therefore dynamic and task-driven, not a fixed hyperparameter.
>
> > L3: The framework relies on online interaction with the environment to correct the erroneous actions making this framework difficult to perform well on offline RL settings.
> >
>
> **A:** HCRMP is intentionally designed to be online, as real-time feedback is core to our 'hint-and-verify' paradigm for safety. We prefer to think of Offline RL as a powerful complement rather than an antithesis. Your question highlights the promising "Offline-to-Online" trend, where offline pre-training provides a strong initialization to accelerate online learning and improve sample efficiency. We thank you for this valuable suggestion and will explore integrating this paradigm into our future work.
>
> > L4: The LLM's role is currently limited to providing semantic scene information.
> >
>
> > L5: The paper does not address how to optimize driving trajectories for human-like behavior, nor does it discuss potential methods for doing so.
> >
>
> **A:** Your suggestions perfectly demonstrate the flexibility of our framework. We are grateful for these ideas, which clarify an exciting direction for our future work. We will detail these potential extensions in the conclusion of our revised manuscript.
>
> Thank you again for your valuable feedback. We will revise the manuscript carefully based on all your comments.

---

> ### Author Response · Authors · 2025-08-07
>
> Dear Reviewer,
>
> I hope this message finds you well.
>
> As the discussion period is nearing its end with less than three days remaining, we wanted to ensure we have addressed all your concerns satisfactorily.
>
> If there are any additional points or feedback you'd like us to consider, please let us know. Your insights are invaluable to us, and we're eager to address any remaining issues to improve our work.
>
> Thank you for your time and effort in reviewing our paper.

---

> ### Comment · Reviewer_L4iX · 2025-08-07
>
> Thanking the authors for addressing the questions.
> I went through the comments and discussions from other reviewers as well and it seems that the authors have tried to carefully address their concerns - many of which were shared across other reviewers.
> - The authors tried to give better clarity as to why this work is not just leveraging the power of LLMs but is a complementary approach to mitigate the negative effects of LLM hallucinations.
>   - Based on their clarification and other experimental results this can be considered a valuable contribution.
> - Though, it would be nice to have more clear objectives defined as well as more exemplary descriptions of the proposed terminologies: LLM-dominated (RL-Assisted LLM Policy Optimization and LLM-instructed RL policy generation), LLM-Assisted RL that seemed to be unclear to other reviewers.
> - I would also recommend to add and highlight the experimental results related to the comparison of:
>   - HCRMP-Attention vs HCRMP,
>   - LLM Hint-based reliance vs LLM strong-reliance
>   - as these are one of the core contributions of the paper.
>   - They seem to be important for this work.
> - Great effort in putting together a real-world deployment experiment to test the research.
>
> Appreciation for the authors was putting efforts in this direction and releasing their work. This work is a good combination of research and practical engineering. I can raise my score given the comments and the clarifications from the authors. I hope to have a better and more comprehensive final version of the paper.
>
> Best wishes! :)

---

> > ### Author Response · Authors · 2025-08-07
> >
> > Dear Reviewer,
> >
> > Thank you so much for your positive and constructive response to our rebuttal. We are very grateful for your time and are very encouraged by your feedback.
> >
> > We also deeply value the concrete suggestions you provided for the final version. We fully agree that they will significantly improve the paper's clarity and impact.
> >
> > As you recommended, we will make sure to:
> >
> > - Provide clearer definitions and more illustrative examples for our proposed terminologies (e.g., LLM-dominated vs. LLM-Hinted).
> > - Prominently feature and highlight the new experimental results, particularly the comparisons between HCRMP-Attention vs. HCRMP and the quantitative analysis of "Hint-based" vs. "Strong-reliance," as you rightly pointed out their importance to our core claims.
> >
> > Thank you once again for your encouraging words and valuable guidance. We are committed to incorporating these improvements into the final manuscript.

---

### Official Review · Reviewer_it5r · 2025-07-04

**Clarity:** 2
**Significance:** 4
**Originality:** 3
**Rating:** 4
**Confidence:** 4

**Summary:**

This paper argues that maintaining relative independence between the LLM and the RL is vital for solving the hallucinations problem.

Reinforcement learning (RL) is <the: you may want to use “a”; there is real-time dynamic programming, and Model-predictive control that also does this; RL is special in that it is sample-based> method for learning optimal policies by maximizing expected returns through interactions with the environment.

From line 52, it is a little confusing. How the LLMs are used with RL for driving? According to the text, “scenario understanding and action response, as illustrated in Figure 1 (a).” How are LLMs used in these scenarios?

Without explaining that clearly, continuing arguing the hallucinations is even more confusing.

RL-Assisted LLM Policy Optimization: I thought it’s the other way around: LLM-Assisted RL Policy Optimization, but this way is also interesting. The review in this section covers a few papers, but it feels still insufficient.

Section 2.2: I see. Here is the other way. So this direction LLM is used to generate intrinsic rewards for RL to improve policy learning. Is this the only way that LLM is used to instruct RL agent? I doubt it.

Figure 2: this is main illustration of the key/framework of the paper. This does not seem to convey easily. It’s better to use some simple example. “On the Expressway”: these texts are helpful but too small and not obvious that they are from LLM.

Fine-grained object-level Analysis: this is not clear what it means. There are no texts. Where and how is the LLM used?

What do you mean by LLM-augmented representations? Texts?

What “prompts” do you mean in the figure?

What is A_int? Value function?Then don’t use A for that. Anyhow, it’s not explained anywhere in the figure or caption.

This is a confusing figure. You don’t want your paper’s idea blurred by a figure that takes an important position and tells not much but confusion.

Section 3.2:
It’s not clear where and how s_t^{llm} is generated? Is it from s_t^{raw}?

Okay A_int is defined here. It’s advantage.

Clipping is a tricky (introduces a few hyperparameters). But Okay. You are following PPO’s way to deal with off-policy sampling ratio. There is a paper that soft-clips the objective without hard clipping.


Experiments are performed on Carla, on Town 2.


References:

Deep Reinforcement Learning for Autonomous Vehicle Intersection Navigation
Distributional Reinforcement Learning for Efficient Exploration: they used Carla too for experiments.

Table 2 and Figure 3 is the main result of experiments.
Put shorthand names like SR in the table caption so it’s easy to find them.

**Questions:**

see above.

**Ethical Concerns:**

["NO or VERY MINOR ethics concerns only"]

**Quality:**

2

**Strengths And Weaknesses:**

LLM-Hinted RL: this has very good result for collision rate. I think you need to do some careful studies into why this is so good. It seems there is no discussion of this in the text.

---

> ### Author Rebuttal · Authors · 2025-07-31
>
> > Q1: Further clarification is requested on how LLMs specifically assist Reinforcement Learning in the aspects of scenario understanding and action response.
> >
>
> **A1:** Thank you for highlighting this narrative shortcoming in our introduction, which we will address in the final version. To clarify, we will detail how, within these methods, an LLM's scenario understanding and action response affect the RL agent, and how hallucinations propagate.
>
> 1. **"RL-Assisted LLM Policy Optimization":** The LLM is the core decision-maker. It primarily leverages its **action response** capability to directly generate high-level driving policies, such as "Longitudinal Driving Decisions" (e.g., accelerate/decelerate) and "Lateral Driving Decisions" (e.g., change lanes). The RL agent then assists in optimizing this LLM-dominated policy through environmental feedback.
>     - If the LLM has a hallucination in its **action response** (e.g., its "Lateral Driving Decisions" capability fails and it outputs a "change lane" command in an unsafe situation), this incorrect command is executed directly, leading to danger.
> 2. **"LLM-Instructed RL Policy Generation":** The LLM primarily acts as an "instructor" leveraging its core **scenario understanding** capability. It is responsible for interpreting and reasoning about complex traffic environments, performing tasks such as "Hazard Identification," "Road Type Comprehension," and "Road sign comprehension." It then translates this high-level understanding into decisive instructions for the RL agent, most commonly by dynamically shaping the reward function.
>     - If the LLM has a hallucination in its **scenario understanding** (e.g., its "Road sign comprehension" capability fails, misinterpreting a clear "Construction Ahead" sign as "road clear"), it will generate a fundamentally flawed reward signal (e.g., continuing to reward "efficient driving" instead of "slowing down"). When optimizing for this contaminated reward, the RL agent is inevitably misled and will learn a dangerous policy, such as driving at high speed into the construction zone.
>
> In both of these modes, the LLM's **scenario understanding** and **action response** are strongly coupled with the final driving decision. If the LLM hallucinates at any stage, the negative impact can propagate directly to vehicle control without a buffer, posing a severe safety hazard.
>
> > Q2: The reviewer is confused by the term "RL-Assisted LLM Policy Optimization" and finds the literature review in that section insufficient. The reviewer also questions whether "LLM-Instructed RL" is limited to only generating intrinsic rewards, suggesting that other methods of instruction likely exist.
> >
>
> **A2:** To ensure the focus of our discussion, we concentrate on studying the methods **most relevant to our work** based on the core mechanism of "how an LLM's output influences and integrates into the RL decision loop," and have not covered all existing interaction paradigms.
>
> We now outline several mainstream paradigms for LLM-instructed RL for autonomous driving.
>
> | Papers & Conferences/Years | Integration Framework (LLM's role) | Collaborative Policy Optimization | LLM Common-Sense Reasoning | LLM makes high-level policy advice | Scenarios Generation | Human-in-the-Loop |
> | --- | --- | --- | --- | --- | --- | --- |
> | Learningflow (arXiv 2025) | **Reward Designer** | **√** | **√** |  |  |  |
> | Lord (WACV 2025) | **Reward Designer** | **√** | **√** |  |  |  |
> | TeLL-Drive (arXiv 2025) | **Decision Provider** | **√** | **√** | **√** |  |  |
> | LaViPlan (arXiv 2025) | **Decision Provider** | **√** | **√** | **√** |  |  |
> | DriveGPT4 (ICLR 2024) | Language Translator | √ |  |  |  | √ |
> | Towards Human-Centric Autonomous Driving (arXiv 2025) | Language Translator | √ |  |  |  | √ |
> | ChatScene  (CVPR 2024) | Generator |  | √ |  | √ |  |
> | CRITICAL (arXiv 2024) | Generator |  | √ |  | √ |  |
> | HCRMP (ours) | Hinter | **√** | **√** |  |  |  |
>
> We will now clarify our reasoning for including or excluding these paradigms from our core discussion:
>
> - **LLM as a Reward Designer:** This approach fully leverages the LLM's powerful common-sense reasoning capabilities for collaborative policy optimization. As this direction aligns with our research, it is a key focus of our discussion.
> - **LLM as a Decision Provider:** This approach positions the LLM as a decision-maker that outputs high-level semantic policies, which are then translated into specific vehicle control signals by an RL module. This is fundamentally different from our RL decision mechanism and is therefore not a core topic of our discussion.
> - **LLM as a Generator:** This approach uses the LLM to enhance the training environment (e.g., by generating long-tail scenarios) rather than for policy optimization. As this is a different technical direction, it is outside the scope of our related work.
> - **LLM as a Language Translator:** This paradigm focuses on parsing human language commands (e.g., "drive faster"). Since our work does not process such commands, we consider its relevance to be limited.
>
> > Q3: The reviewer found Figure 2 to be confusing and raised the following questions and suggestions for revision :
> >
> > 1. Terms like "Fine-grained object-level Analysis" and "LLM-augmented representations" are undefined.
> > 2. The meaning of "prompts" in the figure is unclear.
> > 3. The term Aint is not explained in the figure or its caption.
>
> **A3:** We sincerely apologize for the lack of clarity in Figure 2 and thank you for the detailed feedback.
>
> Our goal for this figure is to intuitively illustrate the HCRMP framework's technical workflow, especially the interaction between its three core modules (ASR, CSA, and SCM) and their influence on the RL agent. Detailed definitions for terms used in the figure, such as "Fine-grained object-level Analysis," are provided in the subsequent subsections of Sec. 3.
>
> To resolve this immediately, we will now clarify the key concepts from the figure:
>
> 1. "LLM-augmented representations": This is the $s_{t}^{llm}$ vector, which is composed of both the "scenario-level" (4-dimensional) and "object-level" (9-dimensional) analyses. As a 13-dimensional semantic hint, it significantly enriches the RL agent's state space.
> 2. "prompts": The "prompts" refer to the structured query texts we input to the LLM. We designed two types of prompts for state augmentation and policy optimization, respectively (detailed prompt design is available in Appendix C).
> 3. $A_{int}$: We apologize for omitting the definition of $A_{int}$ in the caption. $A_{int}$ is our proposed integrated advantage function. As shown in Equation 2, it is the weighted sum of the advantage functions from multiple critics, using the weights ($\lambda_{i}$) generated by the LLM.
>
> For the final version, we will overhaul Figure 2 and its caption for improved clarity. The figure's layout will be redesigned for a more intuitive workflow. Furthermore, the caption will be substantially enriched to define key terms directly and to guide readers step-by-step through our framework's architecture and logic.
>
> > Q4: The paper is unclear on how the $s_{t}^{llm}$ vector is generated and whether it is derived from the $s_{t}^{raw}$ state.
> >
>
> **A4:** Thank you for your question. In Sec. 3.2 (lines 167-174), we described the composition of $s_{t}^{llm}$ conceptually. The semantic information within $s_{t}^{llm}$ is generated via carefully designed prompts fed to the LLM. These detailed prompts, which precisely guide the LLM's "scenario-level abstraction" and "object-level analysis," are fully provided in our Appendix C.1.
>
> $s_{t}^{llm}$ is not generated directly from $s_{t}^{raw}$. They are parallel information streams: $s_{t}^{raw}$ is the agent's direct perceptual input, while $s_{t}^{llm}$ is a semantic supplement from the LLM's reasoning on broader environmental information. The two are then concatenated to form the final state, $\mathcal{S}_{t}$.
>
> We recognize the importance of articulating this key mechanism clearly in the main text. In the final version of our paper, we will **add a more detailed description to Sec. 3.2** to clarify the complete generation process of the $s_{t}^{llm}$ vector from comprehensive environmental information.
>
> > W: The paper presents a significantly improved collision rate for the "LLM-Hinted RL" method but lacks an in-depth analysis to explain the underlying reasons for this strong performance.
> >
>
> **A:**  We appreciate you acknowledging our method's strong performance in reducing the collision rate.
>
> In **Sec. 4.2** of our paper (specifically, lines 245-248 and 263-267), we already provide a preliminary attribution for this phenomenon.
>
> We state that the superior performance is largely due to the CSA module, which reduces collision risk by "dynamically enhances the emphasis on safety" and optimizing the multi-critic coordination strategy. We further attribute this success to the synergy between the ASR and CSA modules, where CSA improves the system's ability to understand complex environments by "extending the state space," ensuring comprehensive driving performance.
>
> To provide stronger empirical evidence as per your suggestion, we will add **Appendix D.4, "Efficacy Analysis of HCRMP,"** to the final paper. This new section will detail further experiments:
>
> - **Critic Weight Trajectory Analysis:** We will plot the critic weights ($\lambda_i$) over time in safety-critical scenarios to show how the CSA module dynamically increases the safety weight ($\lambda_{safety}$) in response to risk.
> - **Counterfactual Intervention Experiment:** We will introduce a "hallucinated" prompt to the LLM during a successful risk-avoidance maneuver. This experiment is designed to prove the robustness of our "loosely-coupled" framework by showing the RL agent can still perform safely.
>
> These additions will rigorously demonstrate the mechanisms behind HCRMP's effectiveness and provide robust support for our central claims.

---

### Comment · Area_Chair_fmtw · 2025-08-04
**Response required.**

Dear reviewers,

As the Author–Reviewer discussion period concludes in a few days, we kindly urge you to read the authors’ rebuttal and respond as soon as possible.

- Please review all author responses and other reviews carefully, and engage in open and constructive dialogue with the authors.

- The authors have addressed comments from all reviewers; each reviewer is therefore expected to respond, so the authors know their rebuttal has been considered.

- We strongly encourage you to post your initial response promptly to allow time for meaningful back-and-forth discussion.

Thank you for your collaboration,
AC

---

> ### Comment · Area_Chair_fmtw · 2025-08-05
> **Reviewer Participation in Discussion Period**
>
> Dear Reviewers,
>
> ACs have been instructed to flag non‑participating reviewers using the Insufficient Review button (Reviewers L4iX and it5r).
> Please ensure that you read the rebuttals and actively contribute to the discussion.
>
> Kindly note that clicking Mandatory Acknowledgement before the discussion period ends does not exempt you from participating. This should only be submitted after you have read the rebuttal and engaged in the discussion.
>
> Best regards,
> AC

---

### Note · Authors · 2025-08-16

Dear Area Chair and Reviewers,

We are immensely grateful for the thorough and constructive feedback provided on our work. In our rebuttal, we believe we have successfully addressed the primary concerns raised. Specifically, we clarified the core contribution of our paper: the introduction of a novel "LLM-Hinted" paradigm, which focuses on safely leveraging LLM capabilities by mitigating hallucination risks, distinct from the "LLM-Dominated" approaches it was initially confused with. To substantiate our claims, we supplemented our original submission with several crucial new experiments, including:

- A quantitative analysis demonstrating the superior robustness of our "Hint-based Reliance" against "Strong Reliance" paradigms when faced with LLM hallucinations;
- A successful real-world vehicle deployment that validates the practical viability of HCRMP and addresses sim-to-real concerns;
- Further ablation studies analyzing prompt sensitivity and comparing our CSA module against learnable alternatives like attention mechanisms.

We are sincerely encouraged by the positive discussion that followed our rebuttal. We are especially grateful to Reviewer it5r for their initial positive assessment and constructive questions. We were also particularly pleased to see that Reviewers L4iX and HcAE found our clarifications and new evidence convincing, leading them to raise their scores. Their follow-up questions, especially the request for a quantitative comparison of reliance types, pushed us to further strengthen the paper's core arguments. **We appreciate that the reviewers recognized our work not merely as an application of LLMs, but as a complementary research direction focused on enhancing the safety and reliability of LLM-RL integration in the critical domain of autonomous driving.**

In light of the valuable feedback, we commit to making the following significant improvements to the final version of our manuscript. We will

- Clarify Core Concepts: Sharpen the "LLM-Hinted" vs. "LLM-Dominated" distinction.
- Improve Visuals: Redesign Figure 2 and its caption for clarity.
- Strengthen Experiments: Integrate new results (reliance analysis, prompt sensitivity) and add normalized metrics (e.g., "Collisions per Mile").

Thank you once again to all reviewers and the Area Chair for your dedicated work and invaluable guidance throughout this process.

---

### Decision · Program_Chairs · 2025-09-17

**Decision:**

Accept (poster)

**Comment:**

This paper explores the integration of large language models (LLMs) with reinforcement learning (RL) to improve motion planning for self-driving cars. While prior approaches often rely heavily on LLM outputs, which can be unreliable due to hallucinations, the authors propose a framework where LLMs provide auxiliary “hints” while the RL agent learns independently, thereby limiting the impact of errors. The system enriches scene understanding, stabilizes decision-making, and bridges the gap between slower LLM guidance and fast driving control. Experiments in CARLA Town2 show higher success rates and fewer collisions than existing LLM–RL baselines.

Reviewers highlighted several strengths. They appreciated the paper’s clear problem motivation around LLM hallucinations, as well as its effort to categorize existing LLM–RL approaches, situating the work in the broader literature. The modular design, comprising semantic augmentation, contextual stabilization, and caching, was seen as well-structured, with the CSA module in particular offering a practical means of stabilizing learning under noisy semantic signals. The analysis of non-hallucination rates of LLMs on driving-related tasks also provides useful insights into current model limitations.

At the same time, reviewers raised important concerns. A recurring issue was a conceptual contradiction: while the paper criticizes over-reliance on LLMs, it still depends heavily on LLM-provided hints, which may themselves suffer from hallucinations and require substantial prompt engineering. Some reviewers also found the methodological presentation confusing, particularly regarding how LLM outputs are generated and incorporated (e.g., figures, definitions, and descriptions of modules). From an evaluation perspective, the lack of key metrics (such as collisions per mile) and the absence of real-world experiments limit confidence in the findings, raising concerns about overfitting to simulation. The degree of novelty was also debated, as similar ideas of using LLM or VLM guidance for RL already exist, and the reported improvements over baselines were seen as modest.

In their rebuttal, the authors clarified that their central contribution lies in introducing an LLM-Hinted paradigm, distinct from LLM-dominated methods, and supplemented their submission with new results. These included a quantitative analysis showing the robustness of hint-based versus strong reliance, a real-world vehicle deployment demonstrating feasibility beyond simulation, and additional ablations on prompt sensitivity and comparisons of the CSA module with alternative designs.

The post-rebuttal discussion was generally positive. One reviewer noted that the clarifications and new experiments effectively addressed earlier concerns, particularly the distinction between hint-based and strong reliance, and expressed satisfaction with the paper. Another reviewer remained unconvinced by this distinction but acknowledged the technical soundness, practical potential and expressed no objection to acceptance.